# Detecting Generated Images by Fitting Natural Image Distributions

**Yonggang Zhang**[1]   **Jun Nie**[2,3]   **Xinmei Tian**[3]   **Mingming Gong**[4,6]   **Kun Zhang**[5,6]   **Bo Han**[2†]
[1]The Hong Kong University of Science and Technology
[2]TMLR Group, Hong Kong Baptist University   [3]University of Science and Technology of China
[4]The University of Melbourne, Australia   [5] Carnegie Mellon University
[6] Mohamed bin Zayed University of Artificial Intelligence

## Abstract

The increasing realism of generated images has raised significant concerns about their potential misuse, necessitating robust detection methods. Current approaches mainly rely on training binary classifiers, which depend heavily on the quantity and quality of available generated images. In this work, we propose a novel framework that exploits geometric differences between the data manifolds of natural and generated images. To exploit this difference, we employ a pair of functions engineered to yield consistent outputs for natural images but divergent outputs for generated ones, leveraging the property that their gradients reside in mutually orthogonal subspaces. This design enables a simple yet effective detection method: an image is identified as generated if a transformation along its data manifold induces a significant change in the loss value of a self-supervised model pre-trained on natural images. Further more, to address diminishing manifold disparities in advanced generative models, we leverage normalizing flows to amplify detectable differences by extruding generated images away from the natural image manifold. Extensive experiments demonstrate the efficacy of this method. Code is available at https://github.com/tmlr-group/ConV.

## 1   INTRODUCTION

Recent advances in generative models have revolutionized image generation, making it possible to create highly realistic images (Rombach et al., 2022; Dhariwal and Nichol, 2021; Karras et al., 2019). While these generative models offer impressive capabilities, they also introduce significant risks, including the proliferation of deepfakes and other manipulated content. The realism achieved by these technologies raises urgent concerns about their potential misuse in sensitive areas like politics and economics. Moreover, if we simply use generated images as part of the training data, the trained model may largely degrade its quality (Shumailov et al., 2024), so it is essential to distinguish between natural images and generated ones. To deal with these potentially dire risks, various generated image detection methods have been developed. In this regard, a common approach is to consider generated image detection as a binary classification task. To train a binary classifier for detecting generated images, current methods typically require to collect numerous natural and generated images to construct a training dataset (Chai et al., 2020; Wang et al., 2020).

Although current methods have achieved exciting success, they often struggle to generalize well to images generated by unknown generative models. To promote the generalization ability on images generated by unknown generative models, one possible approach is to construct a more extensive training dataset by collecting more natural and generated images for training the binary classifier (Jeong et al., 2022; Tan et al., 2024). Besides collecting data, advanced methods propose to

---

[†]Correspondence to Bo Han (bhanml@comp.hkbu.edu.hk).

39th Conference on Neural Information Processing Systems (NeurIPS 2025).

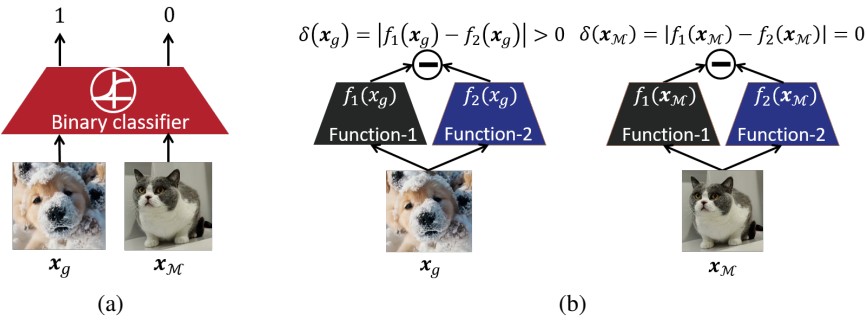

Figure 1: Comparison of (a): the existing framework, and (b): our proposed ConV. The binary classifier in (a) is trained using natural images $\mathbf{x}_{\mathcal{M}}$ and generated images $\mathbf{x}_g$, thereby, its efficacy relies on both the natural and generated data distributions. In contrast, the two functions in (b) are trained on natural data distribution, leading to the advantage of ConV: identifying generated images by fitting the distribution of natural images rather than that of generated images.

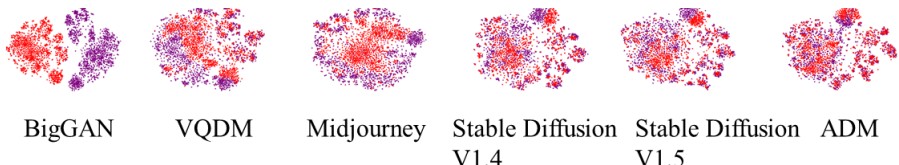

Figure 2: Generated images deviate from natural images' manifold, but the deviation decreases as generative model evolves. Red dots denote the feature representations of natural images, while purple dots represent those of generated images.

introduce pre-trained models as priors to promote the generalization ability. Some works, inspired by the recent success of large models, propose to detect generated images by leveraging features extracted by these large models (Ojha et al., 2023; Liu et al., 2024b), such as CLIP (Radford et al., 2021). Meanwhile, some works propose to leverage the reconstruction capabilities of pre-trained diffusion models (Wang et al., 2023; Ricker et al., 2024). Although these methods have achieved outstanding results, they require a lot of natural and generated images to train a binary classifier, making the current methods computationally intensive. Moreover, sustaining robust detection performance necessitates the continual collection of images generated by the latest generative models, which can be costly or even infeasible due to the inaccessibility of potential models, e.g., Sora OpenAI (2024).

Hence, the major challenge for the existing methods is ensuring that the binary classifier generalizes effectively across diverse unknown generative models. This stems from the fact that these binary classifiers are trained over natural and generated images to distinguish between these two types of images. Thus, the performance of these binary classifiers relies on the diversity of generated data. Unfortunately, it is challenging to determine whether a binary classifier trained over images generated by some diffusion models can generalize to those generated by other models. Their defects of heavy dependence on generated image distribution underscore the necessity of exploring a novel framework for generated image detection, where the detector's performance relies on the natural data distribution rather than the generated image distribution. However, this remains challenging, because the literature has yet to determine whether models training merely on natural images can be leveraged to distinguish between natural and generated images effectively, and if yes, how and why?

To address the challenge, we propose a novel framework for detecting generated images called **con**sistency **v**erification (ConV). Using t-SNE, we visualize the low-dimensional manifold structure of feature representations extracted by DINOv2, which is trained solely on natural images. The embeddings of generated images exhibit distinct patterns compared to those of natural images, as illustrated in Figure 2, supporting the manifold disparities leveraged by our detection framework. To exploit this difference for detecting AI-generated images, as shown in Figure 1, we introduce two functions, aiming to detect generated images by ensuring that the outputs of these functions remain consistent for natural images but exhibit significant inconsistency for generated images. To this end, we establish a principle (see Eq. 6) to design these functions based on our theoretical analysis: outputs of these two functions are the same on the natural distribution while their gradients need to lie within two mutually orthogonal subspaces. This enables a training-free detection approach (see Eq. 12): if an image transformed along its data manifold induces a substantial change in the

loss value of a self-supervised model pre-trained over natural images, it is identified as generated. The advantage of ConV over existing methods is its reliance on fitting the natural data distribution rather than the distribution of generated images. However, as the generative model continues to develop, the deviation between the generated image and the natural image manifold will become smaller, as shown in Figure 2. To address this challenge, we propose actively projecting generated images onto the natural image manifold. The inherent diversity of natural images complicates explicit modeling of this manifold. To overcome this, we employ normalizing flow (Dinh et al., 2017; Kingma and Dhariwal, 2018) to transform the natural image manifold into a Gaussian distribution, enabling precise extrusion of generated images from the natural manifold. This approach significantly enhances detection performance. Comprehensive experiments across various benchmarks for generated image detection demonstrate the effectiveness of the proposed ConV (see Tables 1-7). To further verify the effectiveness of the proposed ConV, we collect images generated by Sora OpenAI (2024) and OpenSora Zheng et al. (2024) and compare ConV with baselines. The experiments demonstrate the efficacy and robustness of ConV against variations in generative models (see Table 2).

We summarize our main contributions as follows:

- We highlight the generalization issue of existing works: it is challenging to determine whether a detector trained over images generated by some diffusion models can generalize to those generated by other models. This motivates a promising direction to explore detectors whose detection ability relies solely on fitting the natural data distribution.

- We propose a novel framework for detecting generated images called **con**sistency **v**erification (ConV). This framework exploits the observed deviation of generated images from natural manifold and detect images by verifying consistency of two functions. The design of these functions is guided by our orthogonality principle. Namely, gradients of these functions need to lie within two mutually orthogonal subspaces (Eq 6). This enables a training-free approach to detecting generated images by leveraging the consistency of a pre-trained self-supervised model on images before and after perturbations along the data manifold.

- To further facilitate the deviation of the generated image, we explicitly extrude generated images out of natural manifold with the aid of normalized flow, enhancing the effectiveness of ConV. Extensive experiments conducted on various standard benchmarks and datasets collected from Sora demonstrate the effectiveness and robustness of the proposed method.

## 2 Consistency Verification

### 2.1 Motivation

Humans can distinguish generated images from natural images through some types of indescribable differences in patterns. Intuitively, humans know that if a natural image captures the same content as a given generated image, the natural image will be different. In contrast, if we degrade natural images along its data manifold, e.g., tiny affine transformation, the degraded natural images are still discriminated as natural images.

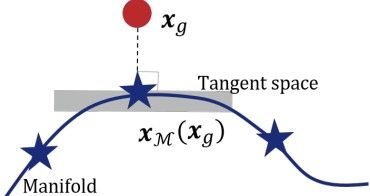

Figure 3: Illustration of projecting a generated image $\mathbf{x}_g$ onto the data manifold $\mathcal{M}$.

To formally characterize this discrepancy, we present the following notations. Let $\mathbf{x} \in \mathcal{X} \subset \mathbb{R}^d$ denote the image, where $d$ denotes the dimension of images. To distinguish, we use $\mathbf{x}_n$ and $\mathbf{x}_g$ to denote the natural and generated image. In particular, for a given generated image $\mathbf{x}_g$, even if it captures similar content to a natural image $\mathbf{x}_n$, humans know they are distinguishable in certain ways. This can be formulated by projecting the generated image $\mathbf{x}_g$ onto the point $\mathbf{x}_{\mathcal{M}(\mathbf{x}_g)}$ on the data manifold $\mathcal{M}$, i.e.,

$$\mathbf{x}_{\mathcal{M}}(\mathbf{x}_g) = \arg \min_{\mathbf{x}' \in \mathcal{M}} d(\mathbf{x}', \mathbf{x}_g), \ \mathbf{x}_{\mathcal{M}}(\mathbf{x}_g) \in \mathcal{M}, \ \mathbf{x}_g \notin \mathcal{M}, \tag{1}$$

where $\mathbf{x}_{\mathcal{M}}(\mathbf{x}_g)$ is the point closest to $\mathbf{x}_g$ on the data manifold of natural images $\mathcal{M}$ and $d$ is a metric. Namely, images on the data manifold $\mathcal{M}$ are considered natural, whereas those deviating from $\mathcal{M}$ are regarded as generated.

In this context, the data manifold perspective provides an intuitive framework for understanding the difference. In particular, transforming natural image $\mathbf{x}_{\mathcal{M}}$ along the local tangent space $\mathcal{T}(\mathbf{x}_{\mathcal{M}})$,

leading to the fact that the degraded images are still on the data manifold. In contrast, even the discrepancy $d(\mathbf{x}_{\mathcal{M}}(\mathbf{x}_g), \mathbf{x}_g)$ is minimal, $\mathbf{x}_g$ is considered as generated, because $\mathbf{x}_g$ departs from the manifold. Intuitively, even a slight discrepancy between $\mathbf{x}_{\mathcal{M}}(\mathbf{x}_g)$ and $\mathbf{x}_g$ allows us to identify the difference between a generated image and the corresponding natural image on the data manifold. Thus, we consider the discrepancy between a generated image and its closest natural image on the data manifold to represent the direction of the fastest departure from the manifold. This means that

$$\mathbf{v}^\top(\mathbf{x}_{\mathcal{M}}(\mathbf{x}_g) - \mathbf{x}_g) = 0, \ \mathbf{v} \in \mathcal{T}(\mathbf{x}_{\mathcal{M}}(\mathbf{x}_g)). \tag{2}$$

This discrepancy inspires us to introduce two functions to detect generated images, where these two functions are related to the tangent space and the space orthogonal to the tangent space, respectively.

## 2.2 Objective

Aligning with the motivation, we introduce a two-function framework for generated image detection. In particular, we propose a consistency verification framework where the two introduced functions are devised to be consistent over natural images and inconsistent over generated images. Namely, this framework detects generated images by verifying the consistency of the two functions. Specifically, let $f_1(\cdot) : \mathbb{R}^d \to \mathbb{R}$ and $f_2(\cdot) : \mathbb{R}^d \to \mathbb{R}$ be the two functions. Then, the inconsistency $|f_1(\cdot) - f_2(\cdot)|$ between these two functions can be employed to detect generated images. Namely, generated images can be detected by $\mathbb{I}(|f_1(\cdot) - f_2(\cdot)| > \alpha)$ with the threshold $\alpha$.

For images on the manifold, we make these two functions consistent by setting

$$\delta(\mathbf{x}_{\mathcal{M}}) = |f_1(\mathbf{x}_{\mathcal{M}}) - f_2(\mathbf{x}_{\mathcal{M}})| = 0, \tag{3}$$

where we denote $\mathbf{x}_{\mathcal{M}}(\mathbf{x}_g)$ as $\mathbf{x}_{\mathcal{M}}$ for simplicity. Then, the objective is to devise the two functions to ensure that the inconsistency over generated images is larger than that over the natural images, i.e., $\delta(\mathbf{x}_g) \geq \delta(\mathbf{x}_{\mathcal{M}})$. In this regard, we show that (with more details in the appendix)

$$\delta(\mathbf{x}_g) \geq |\,|\nabla f_1(\mathbf{x}_{\mathcal{M}})^\top(\mathbf{x}_g - \mathbf{x}_{\mathcal{M}})| - |\nabla f_2(\mathbf{x}_{\mathcal{M}})^\top(\mathbf{x}_g - \mathbf{x}_{\mathcal{M}})|\,| \geq 0 = \delta(\mathbf{x}_{\mathcal{M}}), \tag{4}$$

where equality holds if, and only if, the absolute values of the two quantities are identical. According to Eq. 4, enlarging the difference between these two terms, i.e., $|\nabla f_1(\mathbf{x}_{\mathcal{M}})^\top(\mathbf{x}_g - \mathbf{x}_{\mathcal{M}})|$ and $|\nabla f_2(\mathbf{x}_{\mathcal{M}})^\top(\mathbf{x}_g - \mathbf{x}_{\mathcal{M}})|$ will make the natural and generated images separable. Thus, the objective of consistency verification is to maximize one term and minimize the other term while keeping the output values of these two functions the same. This can be formalized by

$$\min_{f_1, f_2 \in \mathcal{F}} |\nabla f_1(\mathbf{x}_{\mathcal{M}})^\top(\mathbf{x}_g - \mathbf{x}_{\mathcal{M}})| - |\nabla f_2(\mathbf{x}_{\mathcal{M}})^\top(\mathbf{x}_g - \mathbf{x}_{\mathcal{M}})|, \text{s.t. } f_1(\mathbf{x}_{\mathcal{M}}) = f_2(\mathbf{x}_{\mathcal{M}}), \tag{5}$$

where $\mathcal{F}$ denotes a hypothesis space.

However, learning these two functions using Eq. 5 still relies on the generated data, i.e., $\mathbf{x}_g$. To decouple function optimization from the generated data distribution, we leverage orthogonality priors from the motivation 2.1 to provide design principles for these functions. According to the above discussion, one straightforward approach to realizing $f_1$ and $f_2$ is to devise these functions such that their gradients for the input lie in two orthogonal subspaces, i.e., the tangent space and the space orthogonal to the tangent space. This orthogonality principle can be formalized as,

$$\nabla f_1(\mathbf{x}_{\mathcal{M}}) \in \mathcal{O}(\mathbf{x}_{\mathcal{M}}), \ \nabla f_2(\mathbf{x}_{\mathcal{M}}) \in \mathcal{T}(\mathbf{x}_{\mathcal{M}}), \ f_1(\mathbf{x}_{\mathcal{M}}) = f_2(\mathbf{x}_{\mathcal{M}}) \tag{6}$$

where $\mathcal{O}(\mathbf{x}_{\mathcal{M}})$ denotes the subspace orthogonal to the tangent space $\mathcal{T}(\mathbf{x}_{\mathcal{M}})$. Then, we have

$$\delta(\mathbf{x}_g) \geq |\,|\nabla f_1(\mathbf{x}_{\mathcal{M}})^\top \mathbf{p}| - |\nabla f_2(\mathbf{x}_{\mathcal{M}})^\top \mathbf{p}|\,| = |\nabla f_1(\mathbf{x}_{\mathcal{M}})^\top \mathbf{p}| > 0 = \delta(\mathbf{x}_{\mathcal{M}}), \tag{7}$$

where $\mathbf{p} = \mathbf{x}_g - \mathbf{x}_{\mathcal{M}}$ denotes the difference between a generated image and its corresponding point on the data manifold, the equation holds due to the conclusion in Eq. 2, and the inequality holds because the probability that two vectors in the same space are orthogonal is zero. Consequently, the orthogonality principle ensures that these two functions are consistent on natural images, i.e., $f_1(\mathbf{x}_{\mathcal{M}}) = f_2(\mathbf{x}_{\mathcal{M}})$, while inconsistent on generated images, i.e., $|\delta(\mathbf{x}_g)| > |\delta(\mathbf{x}_{\mathcal{M}})| = 0$.

## 2.3 Realization

In this work, we propose a training-free approach to construct these two functions. The reason is twofold: i) our framework allows the training-free construction of these functions, and ii) we aim to

validate the effectiveness of the orthogonality principle without incurring significant energy costs, as fitting the distribution of natural data requires a lot of data and computing power for training.

Well-trained models are typically insensitive to the transformation along the data manifold Simard et al. (1991); Bengio et al. (2013); Rifai et al. (2011). This can be formalized as,

$$(\mathbf{v} - \mathbf{x}_{\mathcal{M}})^{\top} \frac{\partial \ell(\mathbf{x}_{\mathcal{M}})}{\partial \mathbf{x}_{\mathcal{M}}} \approx 0, \quad \mathbf{v} \in \mathcal{T}(\mathbf{x}_{\mathcal{M}}), \tag{8}$$

where $\mathbf{v}$ stands for the point sampled from the tangent space $\mathcal{T}(\mathbf{x}_{\mathcal{M}})$ and $\ell(\cdot)$ is the loss function of a model. This implies that $\frac{\partial \ell(\mathbf{x}_{\mathcal{M}})}{\partial \mathbf{x}_{\mathcal{M}}}$ is orthogonal to the tangent space $\mathcal{T}(\mathbf{x}_{\mathcal{M}})$, which is consistent with the direction $\mathbf{p} = \mathbf{x}_g - \mathbf{x}_{\mathcal{M}}$, as shown in Eq 2. Hence, we propose to realize $f_1(\cdot)$ using a well-trained neural network. This means that both $\frac{\partial \ell(\mathbf{x}_{\mathcal{M}})}{\partial \mathbf{x}_{\mathcal{M}}}$ and $\mathbf{p}$ lies in the subspace orthogonal to tangent space $\mathcal{T}(\mathbf{x}_{\mathcal{M}})$. This is consistent with the principle, i.e., $\nabla f_1(\mathbf{x}_{\mathcal{M}}) \in \mathcal{O}(\mathbf{x})$. We have

$$|\nabla f_1(\mathbf{x}_{\mathcal{M}})^{\top} \mathbf{p}| = |\frac{\partial \ell(\mathbf{x}_{\mathcal{M}})}{\partial \mathbf{x}_{\mathcal{M}}}^{\top} \mathbf{p}| = \left\| \frac{\partial \ell(\mathbf{x}_{\mathcal{M}})}{\partial \mathbf{x}_{\mathcal{M}}} \right\| \|\mathbf{p}\| |\cos(\frac{\partial \ell(\mathbf{x}_{\mathcal{M}})}{\partial \mathbf{x}_{\mathcal{M}}}, \mathbf{p})| > 0, \tag{9}$$

where $\mathbf{p} = \mathbf{x}_g - \mathbf{x}_{\mathcal{M}}$ is the difference between natural and generated images, and the last inequality holds because the probability that two vectors in the same space are orthogonal is zero. We propose to realize $f_1(\cdot)$ using models trained with self-supervised learning, which would avoid reliance on labels used in classification tasks. This is because obtaining the loss value of a classification model requires labels that could be hard to obtain in many practical scenarios.

For the second term, we will realize it using the orthogonality such that $\nabla f_2(\mathbf{x}_{\mathcal{M}}) \in \mathcal{T}(\mathbf{x}_{\mathcal{M}})$ or $|\nabla f_2(\mathbf{x}_{\mathcal{M}})^{\top} \mathbf{p}| = 0$. We achieve this by introducing the local tangent space into $\nabla f_2(\mathbf{x})$. To this end, we propose to realize $f_2$ using a composite function: $f_2 := f_1 \circ h$. This leads to the fact that

$$\nabla f_2(\mathbf{x}_{\mathcal{M}}) = \mathbf{J}_{h(\mathbf{x}_{\mathcal{M}})} \frac{\partial f_1(h(\mathbf{x}_{\mathcal{M}}))}{\partial h(\mathbf{x}_{\mathcal{M}})}, \tag{10}$$

where $\mathbf{J}_{h(\mathbf{x}_{\mathcal{M}})}$ is the Jacobian matrix of $h(\mathbf{x}_{\mathcal{M}})$. If $h(\cdot)$ models the transformation along local data manifold, $\mathbf{J}_{h(\mathbf{x}_{\mathcal{M}})}$ models the tangent space at point $\mathbf{x}_{\mathcal{M}}$. Then, we have

$$\nabla f_2(\mathbf{x}_{\mathcal{M}})^{\top} \mathbf{p} = \frac{\partial f_1(h(\mathbf{x}_{\mathcal{M}}))}{\partial h(\mathbf{x}_{\mathcal{M}})}^{\top} \mathbf{J}_{h(\mathbf{x}_{\mathcal{M}})}^{\top} \mathbf{p} = 0, \tag{11}$$

where $\mathbf{J}_{h(\mathbf{x})}^{\top}$ denotes the tangent space orthogonal to the vector $\mathbf{p} = \mathbf{x}_g - \mathbf{x}_{\mathcal{M}}$, see Eq. 2.

For the last term in the orthogonality principle, we should ensure that $f_1(\mathbf{x}_{\mathcal{M}}) = f_2(\mathbf{x}_{\mathcal{M}}) := f_1(h(\mathbf{x}_{\mathcal{M}}))$. There are numerous approaches to realize $h(\cdot)$. In this regard, we propose to leverage data transformation functions used in the training phase to realize $h(\cdot)$, because self-supervised models are trained to be insensitive to these transformations along local data manifold under various self-supervised learning scenarios (Yu et al., 2023; Jaderberg et al., 2015). Thus, for a given input image $\mathbf{x}$, we can determine whether it is generated by calculating the consistency $\delta(\mathbf{x})$,

$$\delta(\mathbf{x}) = |f_1(\mathbf{x}) - f_1(h(\mathbf{x}))| \begin{cases} = 0, & \mathbf{x} \in \mathcal{M}, \\ > 0, & \mathbf{x} \notin \mathcal{M}. \end{cases} \tag{12}$$

Technically, our training-free realization is equal to verifying the robustness of a pre-trained self-supervised model $f_1(\cdot)$ against the data transformations $h(\cdot)$. Here, $f_1(\cdot)$ merely fits the natural data distribution, avoiding the reliance on the distribution of generated images.

## 2.4 Overview

An overview of the proposed consistency verification is presented in Figure 4. As shown in the figure, our method is training-free and seamlessly deployed in practical scenarios. Specifically, we merely download a neural network pre-trained with a self-supervised learning task over a large-scale dataset. Subsequently, we obtain the loss values of both the original and transformed images. Ultimately, images are identified as generated if the difference between loss values exceeds a predetermined threshold. We can apply multiple random transformations and compute corresponding loss function values if computational resources allow. Intuitively, this would result in more accurate detection performance, which is fortunately consistent with our experiments, see Figure 5.

Negative samples are widely used in self-supervised learning, which could increase the computational cost of generated image detection. Inspired by a recent work Oquab et al. (2024), we calculate the similarity of representation $\mathbf{r} = \phi(\mathbf{x})$, where $\phi(\cdot)$ is the feature extractor of a self-supervised model. The feasibility results from the objective function used in self-supervised learning,

$$\log P(\mathbf{x}) = \log \frac{e^{(\mathbf{r}^\top \mathbf{r}_h / \tau)}}{\sum_{\mathbf{z}_-} e^{(\mathbf{r}^\top \mathbf{r}_- / \tau)} + e^{(\mathbf{r}^\top \mathbf{r}_h / \tau)}} = \log \frac{1}{\sum_{\mathbf{z}_-} e^{(\mathbf{r}^\top \mathbf{r}_- / \tau) - (\mathbf{r}^\top \mathbf{r}_h / \tau)} + 1}, \quad (13)$$

where $\mathbf{r}_h$ is the representation of $h(\mathbf{x})$ and $\mathbf{r}_-$ denotes the representation of negative samples. Thus, we can employ the similarity between representations, i.e., $\mathbf{r}^\top \mathbf{r}_h$, as a surrogate of loss value. This avoids the use of negative samples. Note that applying a softmax function to the representation $\mathbf{r}$ leads to the objective function used in previous works Caron et al. (2021); Oquab et al. (2024). In this context, the high similarity between the representation of images and transformed images means the consistency between functions, i.e., detected as natural images.

## 2.5 Flow-Based Manifold Extrusion

Although generated images, such as those generated by GANs or diffusion models, deviate from the manifold of natural images, this deviation is often subtle, making differentiation challenging. To address this, we aim to actively extrude generated images from the natural image manifold through targeted training, thereby amplifying their separation. However, the challenge comes from the diversity of natural images, which leads to their extremely complex manifolds. This complexity prevents direct modeling of natural man-

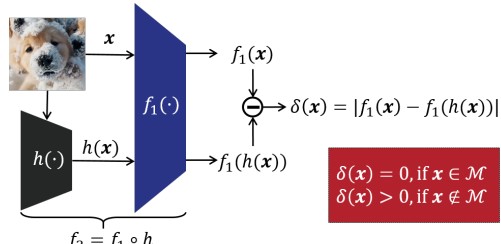

Figure 4: Framework of consistency verification.

ifold. To address this challenge, we propose **F-ConV** to leverage normalizing flows (Dinh et al., 2017) to reshape the natural manifold into a Gaussian distribution, facilitating robust differentiation.

The goal of the normalizing flows is to learn an invertible mapping $f$, which transforms a complex distribution $P_X(v)$ into a simple one $P_Z(z)$: $z = f(v)$ and $v$ is the image feature extracted by the fundamental model: $v = f(x)$ The distribution $P_Z(z)$ is commonly chosen to be a standard Gaussian. Therefore, the objective function for normalizing flow is: Formally, let $f : \mathbb{R}^d \to \mathbb{R}^d$ be the invertible transformation, mapping an input feature $v$ to a latent variable $z = f(v)$, where $z \sim \mathcal{N}(0, I)$. According to the change of variable formula, the log-likelihood of $v$ is:

$$\log p(v) = \log p_z(f(v)) + \log \left| \det \left( \frac{\partial f(v)}{\partial v} \right) \right|, \quad (14)$$

where $\left| \det \left( \frac{\partial f(v)}{\partial v} \right) \right|$ is the determinant of the Jacobian matrix of $f$ at $v$. In order to enforce the deviation of the generated image, we introduce the following loss function:

$$\mathcal{L} = \underbrace{-\mathbb{E}_{v \sim \mathcal{D}_n} \log p(v) + \mathbb{E}_{v \sim \mathcal{D}_g} \log p(v)}_{\text{Shaping Loss}} \underbrace{-\mathbb{E}_{v \sim \mathcal{D}_n} cos(f(v); f(T(v)) + \mathbb{E}_{v \sim \mathcal{D}_g} cos(f(v); f(T(v))}_{\text{Consistency Loss}},$$

$$(15)$$

where $\mathcal{D}_n$ and $\mathcal{D}_g$ denote the distributions of natural and generated images, respectively. $T(v)$ is the feature of transformed image: $T(v) = f(h(x))$. Intuitively, shaping loss pushes the generated images away from natural images' manifold, while the consistency loss amplifies consistency disparity between natural images and generated images. Upon completion of training, we make a judgment of an image by assessing the feature consistency between the original image and its transformed image, as well as its likelihood under the normalizing flow model.

Table 1: Detection performance (%) on ImageNet. Bold numbers are superior results. We compare training methods and training-free methods separately.

| Methods | ADM | | ADMG | | LDM | | DiT | | BigGAN | | GigaGAN | | StyleGAN XL | | RQ-Transformer | | Mask GIT | | Average | |
|---|---|---|---|---|---|---|---|---|---|---|---|---|---|---|---|---|---|---|---|---|
| | AUROC | AP | AUROC | AP | AUROC | AP | AUROC | AP | AUROC | AP | AUROC | AP | AUROC | AP | AUROC | AP | AUROC | AP | AUROC (↑) | AP (↑) |
| Training-based Methods | | | | | | | | | | | | | | | | | | | | |
| CNNspot | 62.25 | 63.13 | 63.28 | 62.27 | 63.16 | 64.81 | 62.85 | 61.16 | 85.71 | 84.93 | 74.85 | 71.45 | 68.41 | 68.67 | 61.83 | 62.91 | 60.98 | 61.69 | 67.04 | 66.78 |
| Ojha | 83.37 | 82.95 | 79.60 | 78.15 | 80.35 | 79.71 | 82.93 | 81.72 | 93.07 | 92.77 | 87.45 | 84.88 | 85.36 | 83.15 | 85.19 | 84.22 | 90.82 | 90.71 | 85.35 | 84.25 |
| DIRE | 51.82 | 50.29 | 53.14 | 52.96 | 52.83 | 51.84 | 54.67 | 55.10 | 51.62 | 50.83 | 50.70 | 50.27 | 50.95 | 51.36 | 55.95 | 54.83 | 52.58 | 52.10 | 52.70 | 52.18 |
| NPR | 85.68 | 80.86 | 84.34 | 79.79 | 91.98 | 86.96 | 86.15 | 81.26 | 89.73 | 84.46 | 82.21 | 78.20 | 84.13 | 78.73 | 80.21 | 73.21 | 89.61 | 84.15 | 86.00 | 80.84 |
| DRCT | 90.26 | 90.07 | 85.74 | 83.85 | 90.24 | 89.88 | 88.27 | 89.06 | 95.87 | 94.99 | 86.89 | 86.12 | 89.11 | 88.39 | 92.38 | 92.41 | 94.44 | 94.47 | 90.36 | 89.92 |
| FatFormer | 91.77 | 90.36 | 83.58 | 83.17 | 92.58 | 92.06 | 86.93 | 85.14 | 98.76 | 98.47 | 97.65 | 98.02 | 97.64 | 97.57 | 96.55 | 95.96 | 97.65 | 97.27 | 93.68 | 93.11 |
| F-ConV | 92.74 | 91.65 | 88.51 | 87.67 | 88.87 | 88.47 | 85.94 | 84.88 | 98.94 | 98.98 | 98.14 | 98.72 | 98.52 | 98.38 | 96.79 | 96.33 | 95.52 | 95.38 | 93.77 | 93.38 |
| Training-free Methods | | | | | | | | | | | | | | | | | | | | |
| AEROBLADA | 55.61 | 54.26 | 61.57 | 56.58 | 62.67 | 60.93 | 85.88 | 87.71 | 44.36 | 45.66 | 47.39 | 48.14 | 47.28 | 48.54 | 67.05 | 67.69 | 48.05 | 48.75 | 57.87 | 57.85 |
| ConV | 88.89 | 86.60 | 82.46 | 79.83 | 78.94 | 75.88 | 75.25 | 70.11 | 92.83 | 92.05 | 91.89 | 90.93 | 92.15 | 91.82 | 93.02 | 91.26 | 88.79 | 87.88 | 87.13 | 85.15 |

Table 2: Detection performance (%) on Sora.

| Models | Methods | | | | | | | | | | | | | | | | | |
|---|---|---|---|---|---|---|---|---|---|---|---|---|---|---|---|---|---|---|
| | CNNspot | | Ojha | | NPR | | DRCT | | DIRE | | FatFormer | | F-ConV | | AEROBLADA | | ConV | |
| | AUROC | AP | AUROC | AP | AUROC | AP | AUROC | AP | AUROC | AP | AUROC | AP | AUROC | AP | AUROC | AP | AUROC | AP |
| Sora | 52.85 | 53.29 | 77.06 | 80.69 | 51.92 | 50.25 | 82.53 | 82.28 | 52.83 | 52.16 | 89.95 | 87.64 | 91.74 | 89.95 | 57.13 | 58.00 | 87.74 | 88.85 |
| Open Sora | 50.14 | 51.38 | 67.05 | 68.67 | 50.25 | 51.84 | 81.79 | 80.11 | 53.66 | 52.98 | 88.76 | 87.99 | 90.16 | 87.38 | 55.79 | 62.37 | 82.84 | 85.24 |
| Average | 51.50 | 52.84 | 72.06 | 74.68 | 51.09 | 51.05 | 82.16 | 81.20 | 53.25 | 52.57 | 89.36 | 87.82 | 90.95 | 88.67 | 56.46 | 60.19 | 85.29 | 87.05 |

## 3 Experiments

### 3.1 Experiment setup

**Datasets.** Following previous work (Chen et al., 2024), we evaluate ConV on several benchmarks: **ImageNet** (Deng et al., 2009), **LSUN-BEDROOM** (Yu et al., 2015), **GenImage** (Zhu et al., 2023b) and **DRCT2M** (Chen et al., 2024). Detailed dataset description can be found in the Appendix G.

Besides these image dataset, current advancements in generative technology have significantly enhanced the realism of synthetic videos (Khachatryan et al., 2023; Blattmann et al., 2023), thereby raising substantial concerns regarding trust in digital media. Moreover, the inaccessibility of their parameters and even their architectures underscores the necessity of verifying the generalization capability of newly proposed detection methods over these generative models. To verify whether the proposed ConV generalizes to these challenging scenarios, we download videos generated by these models and detect images sampled from these videos. Since we currently cannot access the generative model used in Sora (OpenAI, 2024), we gathered several videos and extracted 1,000 frames. Additionally, we generate 100 videos through the open-source OpenSora project (Zheng et al., 2024), extracting 5,000 frames. With these images used as generated images and Laion serving as natural images, we further evaluate ConV's performance and compare it with baselines.

**Baselines and evaluation metrics.** We use training-free and training-based methods as baselines. For training-free methods, we take AEROBLADE (Ricker et al., 2024) as our baseline. For training-based methods, we take DIRE (Wang et al., 2023), CNNspot (Wang et al., 2020), Ojha (Ojha et al., 2023), DRCT (Chen et al., 2024), FatFormer (Liu et al., 2024a) and NPR (Tan et al., 2024) as baselines. For some baselines, we get the results reported in their papers, including Frank (Frank et al.,

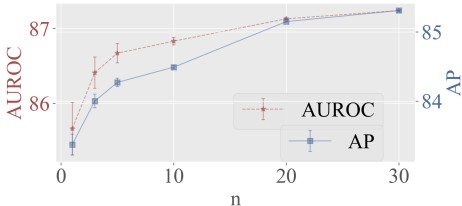

Figure 5: ConV with multiple forward passes.

2020), Durall (Durall et al., 2020), Patchfor (Chai et al., 2020), F3Net (Qian et al., 2020), SelfBland (Shiohara and Yamasaki, 2022), GANDetection (Mandelli et al., 2022), LGrad (Tan et al., 2023), DeiT-S (Touvron et al., 2021), Swin-T (Liu et al., 2021), Spec (Zhang et al., 2019), GenDet (Zhu et al., 2023a) and GramNet (Liu et al., 2020). And following previous works (Ojha et al., 2023; Tan et al., 2024), we mainly use the following metrics: (1) the average precision (AP); (2) the area under the receiver operating characteristic curve (AUROC) and (3) the classification accuracy (ACC).

**Implementation details.** In our experiments, we use the DINOv2 to instantiate $f_1(\cdot)$ and common transformation (details are in Appendix E) to realize $h(\cdot)$. There are four pre-trained DINOv2 models, i.e., ViT-S/14, ViT-B/14, ViT-L/14, and ViT-g/14, achieving exciting AUROC performance on ImageNet benchmark: 62.84, 78.58, 87.13, and 85.97, respectively.

To balance detection performance and efficiency, we use DINOv2 ViT-L/14 in the following experiments. Meanwhile, We leverage data augmentations used in the training phase to realize the

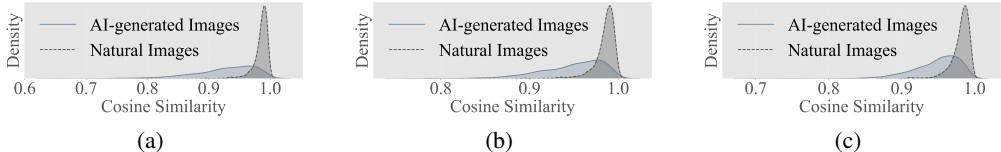

Figure 7: Cosine similarity between features of $\mathbf{x}$ and $h(\mathbf{x})$, where generated images are generated by a) BigGAN, b) ADM, and c) DDPM.

function $h(\cdot)$ in $f_2 = f_1 \circ h$, including geometric augmentation, color jitter, and Gaussian blur. Since data augmentation is randomized, to enhance performance, we apply the function $n$ times to a single test image. As illustrated in Figure 5, increasing $n$ correlates with improved detection performance. However, to maintain detection efficiency, we set $n = 20$ in our experiments. In practical applications, if multiple machines are available, we can leverage parallel processing to implement multiple transformations in a single forward pass to achieve better detection performance. In our experiments, we report the average results under five different random seeds. For F-ConV, we train a RealNVP (Dinh et al., 2017) on the top of DINOv2 ViT-L/14, which consists of 2 coupling blocks with fully connected networks as internal functions. The model is trained using AdamW with a learning rate of 1e-5. More detailed implementation information is provided in Appendix F.

### 3.2 Main Result

**Comparison on public benchmarks.** We conduct comparative experiments across a comprehensive suite of standard benchmarks. As shown in Tables 1, 7, 9 and 8, without training, ConV achieves comparable performance to training methods. And ConV performs better than some of advanced training methods in out-of-distribution generative models. When further extruding the generated image out of natural images' manifold through training, F-ConV achieves the best performance, illustrating the effectiveness of the generalization ability of the proposed method.

**Comparison on Sora.** We further evaluate ConV's performance on videos generated by unknown models. As shown in Table 2, ConV demonstrates the best performance on images generated by these unknown generative models, previous methods. These results highlight the effectiveness and robustness of the proposed ConV.

**Illustration of the effectiveness.** We visualize the features of natural image $\mathbf{x}_n$ and generated image $\mathbf{x}_g$ as well as the features of their augmented versions, i.e., $h(\mathbf{x}_r)$ and $h(\mathbf{x}_g)$. We extract features of $\mathbf{x}_n$, $\mathbf{x}_g$, $h(\mathbf{x}_n)$ and $h(\mathbf{x}_g)$ using DINOv2 and use t-SNE to visualize these features. To avoid the effect of class, all images are sampled from the same class for visualization. As shown in Figure 6, the conclusions are mainly twofold. First, the features of natural ($\mathbf{x}_n$) and augmented ($h(\mathbf{x}_n)$) images can be distinguished from those of generated images and their augmented versions, showing DINOv2's ability to differentiate between natural and generated images. This provides a promising direction to leverage DINOv2 for generated image detection. Second,

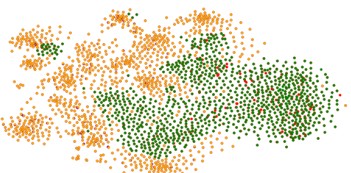

Figure 6: t-SNE visualization of features extracted by DINOv2. Natural image features ($\cdot$ and $\cdot$) remain nearly unchanged after transformations, causing overlap, while generated image features ($\cdot$ and $\cdot$) show notable shifts.

the separation between a generated image and its augmented version in the representation space is more pronounced than that of natural images. The feature of $h(\mathbf{x}_n)$ is similar to that of $\mathbf{x}_n$, i.e., features of $h(\mathbf{x}_n)$ substantially overlap with those of the natural image $\mathbf{x}_n$. In contrast, the features of $h(\mathbf{x}_g)$ generally fail to fully encompass those of the generated images $\mathbf{x}_g$. Aligning with this, ConV effectively distinguishes natural and generated images by calculating feature similarity between the original and augmented images. This is consistent with the conclusion from Figure 7 showing the similarity between features of $\mathbf{x}$ and $h(\mathbf{x})$.

### 3.3 Discussion

When deploying a detector to identify generated images, it is crucial to consider practical environments or even a threat model. Specifically, images are often perturbed in practical scenarios, affecting detection performance. For instance, JPEG compression is a common mechanism due to the spread of images on the Internet. Moreover, AI-generated images may undergo post-processing to evade detection mechanisms. If a detection method is sensitive to some perturbations, the vulnerability

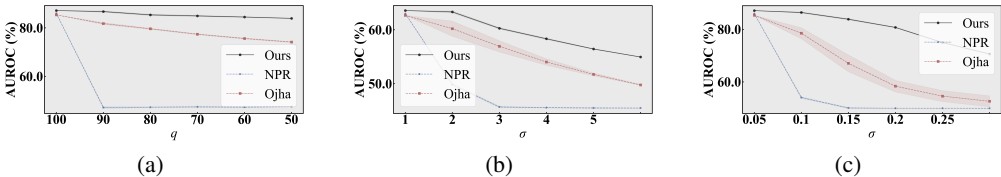

Figure 8: Detection performance under various perturbations: a) JPEG compression, b) Gaussian blur, and c) Gaussian noise.

would limit the applications in many practical scenarios. Thus, robustness to various perturbations is an essential metric in generated image detection. To verify the robustness of the proposed ConV, we process both natural and generated images by introducing some degradation mechanisms. Unless otherwise stated, experiments are conducted on the ImageNet dataset.

Following previous works (Ricker et al., 2024), we evaluate the robustness of ConV in three perturbations, including JPEG compression (with quality $q$), Gaussian blur, and Gaussian noise (both with standard deviation $\sigma$). As shown in Figure 8, ConV achieves the best performance. We find that training-free methods usually show better robustness than training-based methods. Specifically, although NPR achieves promising results on clean images, its performance degrades drastically under perturbations. This may stem from its reliance on the relationship between pixels. Namely, various small perturbations can change its features, causing its performance to degrade drastically. In contrast, ConV leverages the generalization ability of the pre-trained self-supervised model and is robust under various perturbations, making it suitable for a wider range of applications. Besides, we verify the efficacy of the proposed method using more pre-trained models with results in Appendix N. The results demonstrate that our method can be applied for various pre-trained models.

## 4 Related Works

**Generated images detection.** With the rapid advancements in generative models (Brock et al., 2019), the generation of highly realistic images has become increasingly feasible, thereby creating an urgent demand for effective algorithms to detect such generated images. Previous work (Frank et al., 2020) has usually focused on training a specialized binary classification neural network to distinguish between natural and generated images. CNNspot (Wang et al., 2020) finds that with specific data augmentation, a standard image classifier trained on ProGAN is able to generalize to other architectures. However, Ojha (Ojha et al., 2023) shows that the generalizability does not extend to unseen families of generative models. To this end, they propose to train classifiers in CLIP's representation space to obtain stronger generalisability. DIRE (Wang et al., 2023) uses the reconstruction error of an image on a diffusion model to train the classifier. However, training-based approaches often suffer from generalizability issues and high computational costs. To address these limitations, several training-free methods have recently been proposed. AEROBLADE (Tan et al., 2024) performs the detection by calculating the reconstruction error with the autoencoder used in latent diffusion models (Rombach et al., 2022). However, understanding the underlying mechanisms that enable these approaches to perform well on images generated by unknown generative models remains challenging. On the contrary, our method explicitly maps how the generated images are detected. Thus, exhibiting good generalization performance on images generated by unknown models is in line with expectations. Fortunately, our experiments on images generated by Sora and OpenSora provide effective support, see Table 2.

**Manifold learning.** Manifold learning Cayton et al. (2008) assumes that real-world data presented in high dimensional spaces are expected to concentrate in the vicinity of a manifold $\mathcal{M}$ of much lower dimensionality, embedded in high dimensional space. Namely, the probability mass tends to concentrate in regions with significantly lower dimensionality than the original space in which the data resides Bengio et al. (2013). In this context, tangent directions/spaces of the manifold. The tangent space of the manifold changes as the point-of-interest moves on the manifold, as shown in Figure 3. The local tangent space at a point on the manifold can be considered as capturing locally valid transformations, i.e., transformed points are still on the data manifold. Intuitively, a well-trained model is invariant to transformations along the tangent space Simard et al. (1991), which is mathematically equal to the orthogonality between vectors from the tangent space and the gradient of the model's loss with respect to the input, i.e., Eq. 8.

# 5 Conclusion

In this work, we propose ConV, a novel framework for detecting generated images. Unlike existing methods that rely heavily on substantial datasets of natural and generated images, Conv relies solely on the natural image distribution. This is achieved by designing two functions whose outputs exhibit consistency for natural images but significant inconsistency for generated images. Extensive experiments on diverse benchmarks and images generated by a currently inaccessible model, i.e., Sora, have demonstrated ConV's superior performance.

## Acknowledgments

JN and BH were supported by NSFC General Program No. 62376235, RGC Young Collaborative Research Grant No. C2005-24Y, RGC General Research Fund No. 12200725, Guangdong Basic and Applied Basic Research Foundation Nos. 2022A1515011652 and 2024A151501239, and HKBU CSD Departmental Incentive Scheme. XMT was supported by NSFC No. 62222117. MMG was supported by ARC DP240102088 and WIS-MBZUAI 142571. KZ acknowledges the support from NSF Award No. 2229881, AI Institute for Societal Decision Making (AI-SDM), the National Institutes of Health (NIH) under Contract R01HL159805, and grants from Quris AI, Florin Court Capital, and MBZUAI-WIS Joint Program. YGZ was funded by Inno HK Generative AI R&D Center.

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

## A   Social impacts

The proposed AI-generated image detection framework substantially mitigates societal risks stemming from the misuse of generative models. By advancing the capability to detect synthetic media, such as deepfakes, this work strengthens efforts to combat disinformation and enhances trust in digital media across critical domains, including journalism and legal evidence.

## B   Limitation

**Limitation. 1)** Although the proposed orthogonal principle provides an approach for designing various types of functions and its validity is widely supported by extensive empirical studies, we have not provided formal proof of the convergence of the generalization risk within the context of generated image detection. Thus, our future work will focus on establishing the theoretical foundations of the generalization of our approach. **2)** Although we consider a threat model to verify the robustness of detectors, we have not provided an aggressive scenario where generative models are trained to minimize the inconsistency between $f_1$ and $f_2 = f_1 \circ h$. Thus, we will investigate the potential of integrating effective, robust, and efficient detection methods into the training process of generative models to make the generated images more realistic. **3)** Despite numerous empirical studies validating the effectiveness of the proposed ConV, the impact of scaling up the self-supervised model on the performance of detecting generated images remains to be explored since collecting a larger dataset and training an expanded self-supervised model are beyond the scope of this study. Moreover, future work is needed to explore how the performance of ConV will be affected if self-supervised models are trained on generated images. Finally, given the ongoing evolution of generative models, integrating advanced Domain Adaptation techniques (Yang et al., 2023) with ConV appears promising.

## C   Derivation for Inconsistency

Here, we give the detailed derivation of Eq. 4. We expand these two functions at $\mathbf{x} := \mathbf{x}_{\mathcal{M}}(\mathbf{x}_g)$ for a given generated image $\mathbf{x}_g$,

$$f_1(\mathbf{x}_g) = f_1(\mathbf{x}) + \nabla f_1(\mathbf{x})^\top (\mathbf{x}_g - \mathbf{x}), \quad f_2(\mathbf{x}_g) = f_2(\mathbf{x}) + \nabla f_2(\mathbf{x})^\top (\mathbf{x}_g - \mathbf{x}), \qquad (16)$$

where we neglect the higher-order approximation error.

The inconsistency between generated images can be formalized by,

$$\delta(\mathbf{x}_g) = |f_1(\mathbf{x}) - f_2(\mathbf{x}) + (\nabla f_1(\mathbf{x}) - \nabla f_2(\mathbf{x}))^\top (\mathbf{x}_g - \mathbf{x})| = |(\nabla f_1(\mathbf{x}) - \nabla f_2(\mathbf{x}))^\top (\mathbf{x}_g - \mathbf{x})|, \quad (17)$$

where the equation holds because of $\delta(\mathbf{x}) = f_1(\mathbf{x}) - f_2(\mathbf{x}) = 0$. Then, we have

$$\delta(\mathbf{x}_g) = |(\nabla f_1(\mathbf{x}) - \nabla f_2(\mathbf{x}))^\top (\mathbf{x}_g - \mathbf{x})| \geq ||\nabla f_1(\mathbf{x})^\top (\mathbf{x}_g - \mathbf{x})| - |\nabla f_2(\mathbf{x})^\top (\mathbf{x}_g - \mathbf{x})||. \quad (18)$$

## D   SOFTWARE AND HARDWARE

We use python 3.8.16 and Pytorch 1.12.1, and seveal NVIDIA GeForce RTX-3090 GPU and NVIDIA GeForce RTX-4090 GPU.

## E   Details of transformations

We follow the data augmentation strategy used when training DINOv2 with a combination of HorizontalFlip, ColorJitter, and GaussianBlur. For ColorJitter, brightness, contrast, saturation, and hue are randomly adjusted with a factor in the ranges of [0.88,1.12],[0.88,1.12],[0.94,1.06], and[0.97,1.03], respectively. For GaussianBlur, the kernel size is set to 9×9, and the variance is randomly selected in [0.7,1].

## F   Implementation Details

To maintain detection efficiency, we set $n = 20$ in our experiments. For F-ConV, we train a RealNVP (Dinh et al., 2017) on the top of DINOv2 ViT-L/14, which consists of 2 coupling blocks

Table 3: AI-generated image detection performance on ImageNet.

| Methods | ADM | | ADMG | | LDM | | DiT | | BigGAN | | GigaGAN (Models) | | StyleGAN XL | | RQ-Transformer | | Mask GIT | | Average | |
|---|---|---|---|---|---|---|---|---|---|---|---|---|---|---|---|---|---|---|---|---|
| | AUROC | AP | AUROC | AP | AUROC | AP | AUROC | AP | AUROC | AP | AUROC | AP | AUROC | AP | AUROC | AP | AUROC | AP | AUROC ($\uparrow$) | AP ($\uparrow$) |
| Random rotation (-90-90 degrees) | 74.43 | 75.23 | 67.44 | 66.45 | 65.60 | 65.12 | 65.47 | 65.71 | 75.20 | 76.89 | 71.72 | 74.41 | 74.66 | 77.13 | 76.36 | 77.62 | 71.21 | 72.95 | 71.34 | 72.39 |
| Random rotation (-45-45 degrees) | 79.91 | 79.12 | 71.61 | 68.80 | 69.65 | 66.87 | 70.03 | 68.12 | 82.11 | 81.95 | 79.21 | 79.65 | 83.09 | 83.58 | 82.79 | 82.03 | 77.91 | 77.54 | 77.37 | 76.41 |

with fully connected networks as internal functions. The model is optimized using the AdamW optimizer with a learning rate of $1 \times 10^{-5}$, $\beta_1 = 0.9$, $\beta_2 = 0.99$, and a weight decay of 0.01. Following CNNspot (Wang et al., 2020), data augmentation techniques, including JPEG compression and Gaussian blur, are applied to enhance model robustness. For the ImageNet and LSUN-Bedroom benchmarks, the ProGAN dataset is used as the training set. For the GenImage benchmark, the SDv1.4 dataset is employed as trainind set and the SDv2 dataset serves as the training set for the DRCT-2M benchmark. During testing, to ensure unbiased classification accuracy and mitigate biases from manually selected thresholds, we follow (Ojha et al., 2023) by automatically determining the optimal threshold. This threshold is selected to maximize the separation between natural and synthetic images based on their classification scores. An alternative approach involves optimizing the threshold on a small validation set. This method's performance is sensitive to the validation set's characteristics, such as its size and distributional representativeness. However, in real-world applications, the methods for determining thresholds may vary. For instance, to prioritize the detection of generated images, administrators may set a higher threshold, classifying only samples with similarity scores approaching 1 as natural images. This is consistent with the motivation of leveraging AUROC and AP as the main metric for evaluation.

# G  Details of Datasets

**IMAGENET.** The natural images and generated images can be obtained at `https://github.com/layer6ai-labs/dgm-eval`. The images are provided by (Stein et al., 2023). The resolution of natural images and generated images are $256 \times 256$. The generated images include: ADM, ADMG, BigGAN, DiT-XL-2, GigaGAN, LDM, StyleGAN-XL, RQ-Transformer, Mask-GIT

**LSUN-BEDROOM.** The natural images and generated images can be obtained at `https://github.com/layer6ai-labs/dgm-eval`. The images are provided by (Stein et al., 2023). The resolution of natural images and generated images are $256 \times 256$. We crop the image randomly to $224 \times 224$ resolution. The generated images include: ADM, DDPM, iDDPM, StyleGAN, Diffusion-Projected GAN, Projected GAN, Unleashing Transformers.

**GenImage.** The natural images and generated images can be obtained at `https://github.com/GenImage-Dataset/GenImage`. The images are provided by (Zhu et al., 2023b). The natural images come from ImageNet, and different images have different resolutions. The generative model includes Midjourney, SD V1.4, SD V1.5, ADM, GLIDE, Wukong, VQDM, and BigGAN.

**DRCT-2M.** The natural images of DRCT-2M come from CoCo and can be obtained from `https://cocodataset.org/#download`. AI-generated images of DRCT-2M can be obtained from `https://modelscope.cn/datasets/BokingChen/DRCT-2M/files`, which are provided by (Chen et al., 2024). The generative model includes LDM, SDv1.4, SDv1.5, SDv2, SDXL, SDXL-Refiner, SD-Turbo, SDXL-Turbo, LCM-SDv1.5, LCM-SDXL, SDv1-Ctrl, SDv2-Ctrl, SDXL-Ctrl, SDv1-DR, SDv2-DR, SDXL-DR.

# H  Results of using other data transformations.

In our experiments, we leverage data augmentations used in the training phase, including geometric augmentations, color jitter, and Gaussian blur. We further conduct comparison experiments using data augmentations which is not used during training, such as random rotation. The experiments are conducted on the ImageNet

Table 4: Ablation studies on F-ConV.

| Method | AUROC | AP |
|---|---|---|
| F-ConV | 93.77 | 93.38 |
| w/o Shaping Loss | 89.17 | 87.89 |
| w/o Consistency Loss | 92.54 | 91.66 |

benchmark. As shown Table 3, using data transformations not seen during training does not result in

Table 6: AI-generated image detection performance on ImageNet.

| Methods | ADM | | ADMG | | LDM | | DiT | | BigGAN | | GigaGAN | | StyleGAN XL | | RQ-Transformer | | Mask GIT | | Average | |
|---|---|---|---|---|---|---|---|---|---|---|---|---|---|---|---|---|---|---|---|---|
| | AUROC | AP | AUROC | AP | AUROC | AP | AUROC | AP | AUROC | AP | AUROC | AP | AUROC | AP | AUROC | AP | AUROC | AP | AUROC (↑) | AP (↑) |
| Training-based Methods | | | | | | | | | | | | | | | | | | | | |
| CNNspot | 62.25 | 63.13 | 63.28 | 62.27 | 63.16 | 64.81 | 62.85 | 61.16 | 85.71 | 84.93 | 74.85 | 71.45 | 68.41 | 68.67 | 61.83 | 62.91 | 60.98 | 61.69 | 67.04 | 66.78 |
| Ojha | 83.37 | 82.95 | 79.60 | 78.15 | 80.35 | 79.71 | 82.93 | 81.72 | **93.07** | **92.77** | 87.45 | 84.88 | 85.36 | 83.15 | 85.19 | 84.22 | **90.82** | **90.71** | 85.35 | 84.25 |
| DIRE | 51.82 | 50.29 | 53.14 | 52.96 | 52.83 | 51.84 | 54.67 | 55.10 | 51.62 | 50.83 | 50.70 | 50.27 | 50.95 | 51.36 | 55.95 | 54.83 | 52.58 | 52.10 | 52.70 | 52.18 |
| NPR | 85.68 | 80.86 | **84.34** | 79.79 | **91.98** | **86.96** | **86.15** | 81.26 | 89.73 | 84.46 | 82.21 | 78.20 | 84.13 | 78.73 | 80.21 | 73.21 | 89.61 | 84.15 | 86.00 | 80.84 |
| Training-free Methods | | | | | | | | | | | | | | | | | | | | |
| AEROBLADA | 55.61 | 54.26 | 61.57 | 56.58 | 62.67 | 60.93 | 85.88 | **87.71** | 44.36 | 45.66 | 47.39 | 48.14 | 47.28 | 48.54 | 67.05 | 67.69 | 48.05 | 48.75 | 57.87 | 57.85 |
| ConV-DINOv2 | **88.89** | **86.60** | 82.46 | **79.83** | 78.94 | 75.88 | 75.25 | 70.11 | 92.83 | 92.05 | **91.89** | **90.93** | **92.15** | **91.82** | **93.02** | **91.26** | 88.79 | 87.88 | **87.13** | **85.15** |
| ConV-CLIP-unimodal | 76.64 | 76.52 | 69.36 | 68.86 | 70.29 | 69.73 | 70.03 | 69.73 | 76.59 | 79.27 | 72.97 | 73.05 | 70.82 | 70.35 | 77.27 | 77.49 | 72.95 | 73.20 | 72.99 | 72.98 |
| ConV-CLIP-multimodal | 80.76 | 79.77 | 72.31 | 71.21 | 72.03 | 71.22 | 72.73 | 72.12 | 80.73 | 76.60 | 79.59 | 77.47 | 77.46 | 75.17 | 80.83 | 78.86 | 74.34 | 70.45 | 76.75 | 74.76 |

good detection performance. Since the rotations were not used for data augmentation during training, using them to perform ConV during testing could not achieve good detection performance.

# I  Ablation studies on the transformation functions

As shown in Table 5, we conduct additional ablation studies evaluating the effectiveness of various transformation functions. The results indicate that the three transformations examined exhibit comparable performance.

# J  Ablation studies on F-ConV

As shown in Table 4, we perform ablation experiments on the two loss functions used in F-ConV. The results validate the effectiveness of our approach.

Table 5: The effect of transformation functions.

| Model | AUROC | AP |
|---|---|---|
| ConV | 87.13 | 85.15 |
| w/o HorizontalFlip | 86.55 | 84.19 |
| w/o ColorJitter | 85.97 | 83.74 |
| w/o GaussianBlur | 85.49 | 82.83 |

# K  Analysis of Failure Cases

As shown in Figure 9, we demonstrate some failure cases of ConV. We compute the original features and transformed features on highly realistic generated images. It can be observed that the features of these highly realistic generated images also remain virtually unchanged, leading the model to incorrectly classify them as natural images.

# L  Results on CLIP

In our paper, we use DINOv2 for all of our experiments. We further use CLIP for comparison experiments. We note that the authors only used randomly crop as data augmentation when training CLIP. Therefore, when implementing ConV with CLIP, we also only use random crop. As shown in Table 6, using CLIP to implement ConV does not achieve good performance. We speculate that this difference comes from the training methodology. CLIP learns features using image captions as supervision, which may make the features more focused on semantic information, whereas DINOv2 learns features only from images, which makes it more focused on the images themselves, and thus better able to capture the subtle differences between the natural image and the generated image. In addition to this, the fact that CLIP only uses random crop as data augmentation may also contribute to the poor performance of ConV.

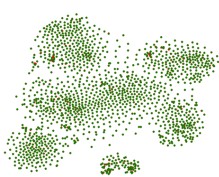

Figure 9: ConV fails on highly realistic generated images.

The results show that our method performs relatively worse when using CLIP. To overcome this limitation, we revisit our methodology, i.e., verifying the consistency between outputs of two functions.

As shown in Eq. (13), we derive the cosine similarity metric between image features from a self-supervised learning objective function. However, CLIP employs a different objective function, namely, calculating the similarity between text and image features. Thus, the proposed cosine similarity

between image features may not be a good realization of these two functions' output, limiting the generalization capability for generated image detection. We conjecture the difference between the projection of the visual features of the original image and the visual features of the transformed image on their corresponding text features would be a good metric. The reason is as follows: The function to calculate the similarity between text and image features can be regarded as a function. Thus, we should calculate the difference in inter-modality similarity rather than the similarity between original and transformed images. To verify the point, we conduct experiments using the corrected realization of $f_1$ and $f_2$, i.e., a corrected metric to verify the consistency. The results below show that the modified approach outperforms the original metric.

These results show that using the modified metric for detection greatly improves the performance of CLIP-based methods model, achieving performance comparable with Dinov2-based methods. Hence, we believe our work provides a novel approach to calculating the difference between two functions without focusing on the differences in similarities between two image features.

## M  Results on GenImage, DRCT-2M and LSUN-BEDROOM

As shown in Table 7, Table 9 and Table 8, our method achieves good performance on GenImage, DRCT-2M and LSUN-BEDROOM, confirming the robustness of the proposed method.

Table 7: AI-generated image detection performance (ACC, %) on GenImage.

| | Models | | | | | | | | |
|---|---|---|---|---|---|---|---|---|---|
| Methods | Midjourney | SD V1.4 | SD V1.5 | ADM | GLIDE | Wukong | VQDM | BigGAN | Average |
| | Training Methods | | | | | | | | |
| ResNet-50 | 54.9 | 99.9 | 99.7 | 53.5 | 61.9 | 98.2 | 56.6 | 52.0 | 72.1 |
| DeiT-S | 55.6 | 99.9 | 99.8 | 49.8 | 58.1 | 98.9 | 56.9 | 53.5 | 71.6 |
| Swin-T | 62.1 | 99.9 | 99.8 | 49.8 | 67.6 | 99.1 | 62.3 | 57.6 | 74.8 |
| CNNspot | 52.8 | 96.3 | 95.9 | 50.1 | 39.8 | 78.6 | 53.4 | 46.8 | 64.2 |
| Spec | 52.0 | 99.4 | 99.2 | 49.7 | 49.8 | 94.8 | 55.6 | 49.8 | 68.8 |
| F3Net | 50.1 | 99.9 | 99.9 | 49.9 | 50.00 | 99.9 | 49.9 | 49.9 | 68.7 |
| GramNet | 54.2 | 99.2 | 99.1 | 50.3 | 54.6 | 98.9 | 50.8 | 51.7 | 69.9 |
| DIRE | 60.2 | 99.9 | 99.8 | 50.9 | 55.0 | 99.2 | 50.1 | 50.2 | 70.7 |
| Ojha | 73.2 | 84.2 | 84.0 | 55.2 | 76.9 | 75.6 | 56.9 | 80.3 | 73.3 |
| NPR | 81.0 | 98.2 | 97.9 | 76.9 | 89.8 | 96.9 | 84.1 | 84.2 | 88.6 |
| FatFormer | 92.7 | 100.0 | 99.9 | 75.9 | 88.0 | 99.9 | 98.8 | 55.8 | 88.9 |
| GenDet | 89.6 | 96.1 | 96.1 | 58.0 | 78.4 | 92.8 | 66.5 | 75.0 | 81.6 |
| DRCT | 91.5 | 95.0 | 94.4 | 79.4 | 89.1 | 94.6 | 90.0 | 81.6 | 89.4 |
| F-Conv | 89.3 | 98.8 | 98.5 | 74.9 | 89.3 | 95.6 | 86.7 | 87.6 | **90.1** |
| | Training-free Methods | | | | | | | | |
| AEROBLADE | 80.3 | 87.5 | 86.8 | 67.2 | 81.5 | 83.7 | 51.1 | 52.5 | 73.83 |
| ConV | 85.13 | 76.74 | 74.53 | 73.80 | 72.97 | 80.00 | 87.57 | 89.94 | **80.08** |

Table 8: Detection performance (%) on LSUN-Bedroom.

| | Models | | | | | | | | | | | | | Average | |
|---|---|---|---|---|---|---|---|---|---|---|---|---|---|---|---|
| Methods | ADM | | DDPM | | iDDPM | | Diffusion GAN | | Projected GAN | | StyleGAN | | Unleashing Transformer | | | |
| | AUROC | AP | AUROC | AP | AUROC | AP | AUROC | AP | AUROC | AP | AUROC | AP | AUROC | AP | AUROC (↑) | AP (↑) |
| CNNspot | 64.83 | 64.24 | 79.04 | 80.58 | 76.95 | 76.28 | 88.45 | 87.19 | 90.80 | 89.94 | 95.17 | 94.94 | 93.42 | 93.11 | 84.09 | 83.75 |
| Ojha | 71.26 | 70.95 | 79.26 | 78.27 | 74.80 | 73.46 | 84.56 | 82.91 | 82.00 | 78.42 | 81.22 | 78.08 | 83.58 | 83.48 | 79.53 | 77.94 |
| DIRE | 57.19 | 56.85 | 61.91 | 61.35 | 59.82 | 58.29 | 53.18 | 53.48 | 55.35 | 54.93 | 57.66 | 56.90 | 67.92 | 68.33 | 59.00 | 58.59 |
| NPR | 75.43 | 72.60 | 91.42 | 90.89 | 89.49 | 88.25 | 76.17 | 74.19 | 75.07 | 74.59 | 68.82 | 63.53 | 84.39 | 83.67 | 80.11 | 78.25 |
| F-ConV | 76.59 | 74.40 | 93.53 | 92.16 | 88.90 | 86.85 | 98.10 | 98.03 | 97.93 | 97.81 | 91.63 | 90.16 | 97.31 | 96.91 | **92.00** | **90.91** |
| AEROBLADA | 57.05 | 58.37 | 61.57 | 61.49 | 59.82 | 61.06 | 47.12 | 48.25 | 45.98 | 46.15 | 45.63 | 47.06 | 59.71 | 57.34 | 53.85 | 54.25 |
| ConV | 73.71 | 71.52 | 87.74 | 86.59 | 82.96 | 81.79 | 93.79 | 93.87 | 94.73 | 94.74 | 84.10 | 82.35 | 93.75 | 93.51 | **87.25** | **86.34** |

Table 9: AI-generated image detection performance (ACC, %) on DRCT-2M.

| Method | SD Variants | | | | | | Turbo Variants | | LCM Variants | | ControlNet Variants | | | DR Variants | | | Avg. |
|---|---|---|---|---|---|---|---|---|---|---|---|---|---|---|---|---|---|
| | LDM | SDv1.4 | SDv1.5 | SDv2 | SDXL | SDXL-Refiner | SD-Turbo | SDXL-Turbo | LCM-SDv1.5 | LCM-SDXL | SDv1-Ctrl | SDv2-Ctrl | SDXL-Ctrl | SDv1-DR | SDv2-DR | SDXL-DR | |
| CNNspot | 99.87 | 99.91 | 99.90 | 97.63 | 66.25 | 86.55 | 86.15 | 72.42 | 98.26 | 61.72 | 97.96 | 85.89 | 82.94 | 60.93 | 51.41 | 50.28 | 81.12 |
| F3Net | 99.85 | 99.78 | 99.79 | 88.60 | 55.85 | 87.37 | 63.29 | 63.66 | 97.39 | 54.98 | 97.98 | 72.39 | 81.99 | 65.42 | 50.39 | 50.27 | 71.13 |
| CLIP/RN50 | 99.00 | 99.99 | 99.96 | 94.61 | 62.08 | 91.43 | 84.40 | 64.40 | 98.97 | 57.43 | 99.74 | 80.69 | 82.03 | 65.83 | 50.67 | 50.47 | 80.05 |
| GramNet | 99.40 | 99.01 | 98.84 | 95.30 | 62.63 | 80.68 | 71.19 | 69.32 | 93.05 | 57.02 | 89.97 | 75.55 | 82.68 | 51.23 | 50.01 | 50.08 | 76.62 |
| De-fake | 92.1 | 95.53 | 99.51 | 89.65 | 64.02 | 69.24 | 92.00 | 93.93 | 99.13 | 70.89 | 58.98 | 62.34 | 66.66 | 50.12 | 50.16 | 50.00 | 75.52 |
| Conv-B | 99.97 | 100.0 | 99.97 | 95.84 | 64.44 | 82.00 | 60.75 | 99.27 | 99.27 | 62.33 | 99.80 | 83.40 | 73.28 | 61.65 | 51.79 | 50.41 | 79.11 |
| Ojha | 98.30 | 96.22 | 96.33 | 93.83 | 91.01 | 93.91 | 86.38 | 85.92 | 90.44 | 89.99 | 90.41 | 81.06 | 89.06 | 51.96 | 51.03 | 50.46 | 83.46 |
| DIRE | 54.62 | 75.89 | 76.04 | 99.87 | 59.90 | 93.08 | 97.55 | 87.29 | 72.53 | 67.85 | 99.69 | 64.40 | 64.40 | 49.96 | 52.48 | 49.92 | 72.55 |
| DRCT | 94.45 | 94.35 | 94.24 | 95.05 | 96.41 | 95.38 | 94.81 | 94.48 | 91.66 | 95.54 | 93.86 | 93.50 | 93.54 | 84.34 | 83.20 | 67.61 | 91.35 |
| FatFormer | 96.52 | 95.31 | 93.27 | 91.99 | 92.87 | 91.78 | 88.15 | 87.48 | 92.82 | 91.76 | 90.28 | 86.99 | 88.19 | 65.92 | 60.15 | 55.13 | 85.53 |
| ConV | 88.12 | 74.75 | 73.17 | 79.13 | 82.10 | 89.53 | 78.25 | 77.92 | 77.15 | 86.37 | 77.67 | 77.85 | 86.73 | 62.79 | 60.18 | 57.83 | 76.84 |
| F-ConV | 99.07 | 98.38 | 98.84 | 99.05 | 98.75 | 99.50 | 98.29 | 97.66 | 98.58 | 98.99 | 98.56 | 98.01 | 97.95 | 69.74 | 65.87 | 64.88 | **92.63** |

Table 10: generated image detection performance with different pre-trained models.

| Methods | MoCo | | SwAV | | Models
DINO | | CLIP | | DINOv2 | |
|---|---|---|---|---|---|---|---|---|---|---|
| | AUROC | AP | AUROC | AP | AUROC | AP | AUROC | AP | AUROC | AP |
| ConV | 68.43 | 67.65 | 74.71 | 73.48 | 71.91 | 69.46 | 72.99 | 72.98 | 87.13 | 85.15 |

# N    Result on more pre-trained models

Besides CLIP, we conduct experiments using the MoCo (He et al., 2020), SwAV (Caron et al., 2020), and DINO (Caron et al., 2021). The results are reported in Table 10. These results show that our method can be applied to various backbones.

Table 11: Experimental results on Flux.

| Method | AUROC | AP |
|---|---|---|
| ConV | 87.38 | 89.19 |
| F-ConV | 90.18 | 90.65 |

# O    Experimental results on Flux

To further assess the generalization capabilities of ConV, we evaluate its performance against the advanced FLUX.1 [dev] generative model (Batifol et al., 2025), which was not encountered during training. For this analysis, a new benchmark dataset is constructed, comprising 6,000 images generated by FLUX.1 [dev] and an equal number of natural images sampled from ImageNet. As presented in Table 11, ConV demonstrates robust performance on this unseen model, underscoring the strong cross-model generalizability of our approach.

# P    Comparison with Linear Classifiers

To further demonstrate the effectiveness of our approach, we additionally train a binary classifier on the DINOv2 embeddings using the same training set. As shown in Table 12, the performance of the directly trained binary classifier falls short of that achieved by F-ConV, highlighting the advantage of our manifold-based approach.

Table 12: Comparison with Linear Classifiers.

| Method | AUROC | AP |
|---|---|---|
| ConV | 87.38 | 89.19 |
| F-ConV | 90.18 | 90.65 |
| Linear classification | 87.83 | 86.49 |

