# OpenReview forum: "Detecting Generated Images by Fitting Natural Image Distributions"
_NeurIPS.cc/2025/Conference — NeurIPS 2025 spotlight_

### Official Review · Reviewer_QsoV · 2025-06-30

**Clarity:** 2
**Significance:** 3
**Originality:** 3
**Rating:** 4
**Confidence:** 4

**Summary:**

This paper proposes ConV, a framework for detecting synthetic images by exploiting geometric differences between the data manifolds of natural and generated images. Specifically, the method measures feature similarity between an image and its augmented version using a self-supervised model pretrained on natural images (e.g., DINOv2). If the difference exceeds a certain threshold, the image is classified as synthetic. To further enhance detection performance, the authors proposed F-ConV that introduces a flow-based extension that uses normalizing flows to push synthetic images away from the natural manifold. Experiments on standard benchmarks and Sora-generated datasets demonstrate the effectiveness of both ConV and F-ConV.

**Questions:**

1. Regarding Figure 2: How are the embeddings generated; What do the “natural images” and “synthetic images” refer to in this visualization?
2. Can the proposed method be extended to detect AI-edited images (e.g., real images with minor synthetic modifications)?
3. *Page 4, Line 134:* The text states: “we denote $x_M(x_g)$ as $x_M$ for simplicity.” Should here be  $\texttt{x}_\mathcal{M}(\texttt{x}_n)$ referring to the natural image instead?

**Ethical Concerns:**

["NO or VERY MINOR ethics concerns only"]

**Final Justification:**

While I still maintain my view that, based on the current implementation, augmentation invariance for natural images in SSL models could serve as an equally appropriate and perhaps more intuitive theoretical foundation (in the sense that one could develop a paper with a similar implementation/experiment but framed under this alternative motivation, and potentially making the theoretical justification more straightforward), most of my concerns regarding the proposed motivation have been addressed. I have accordingly adjusted my rating upward. I also appreciate the authors’ timely and thorough engagement throughout our discussion.

**Limitations:**

Please refer to the **Weaknesses** section above for a discussion of key limitations.

**Quality:**

3

**Strengths And Weaknesses:**

**Strengths:**

1. The paper is well written and clearly structured.
2. Experimental results are thorough, with comparisons against a wide range of strong and up-to-date baselines.
3. The proposed method is conceptually novel and computationally efficient, particularly in its training-free form, which performs competitively with existing training-based approaches for detecting synthetic images from certain generative models.

**Weaknesses**

1. **Gap Between Motivation and Implementation:** While the paper is motivated by geometric differences in data manifolds and an orthogonality principle between functions $f_1$ and $f_2$, the implementation primarily relies on the augmentation invariance learned during large-scale self-supervised training on natural images—an advantage not shared by unseen synthetic images. This central discriminative factor is not sufficiently discussed in the motivation (Sections 2.1–2.3). Moreover, the theoretical formulation assumes both $f_1$ and $f_2$ are differentiable, yet $f_2$ includes a random augmentation operator, which is not explicitly differentiable. These discrepancies suggest a deviation between the theoretical premise and the practical implementation.

2. **Insufficient Validation of Flow-Based Extrusion (F-ConV):** The benefits of using normalizing flows to extrude generated images away from the natural manifold are not fully validated. For instance, adding a supervised classifier on top of DINOv2 embeddings could serve as a baseline to isolate and better understand the contribution of the flow-based component.
3. **Thresholding Details:** The threshold that determines whether an image is synthetic is a critical component of the proposed method. While briefly mentioned in Appendix (Page 14, Line 514), more detail/discussion is needed regarding how this threshold is chosen and whether it generalizes.
4. **Performance Variability Across Generative Models:** As shown in Tables 1 and 5, the proposed ConV/F-ConV do not consistently outperform all baselines. For images generated by models such as DiT, Stable Diffusion v1.4/1.5, and GLIDE, performance lags behind other methods by a notable margin. A deeper analysis of these failure cases could yield valuable insights into the limitations of the approach and its sensitivity to different generative model architectures or styles.

---

> ### Author Rebuttal · Authors · 2025-07-31
>
> We are truly grateful for the reviewers' dedicated efforts. The valuable comments greatly improve the quality of our work. We provide the following answers to the reviewers' concerns.
>
>
> > Q1. Gap Between Motivation and Implementation, and differentiable issue.
>
> Inresponse to your valuable comment, we have added the following details to our revision to highlight the derivation process of our method.
>
> Our method is based on the proposed orthogonality principle, i.e., Eq. 6. Subsequently, we introduce Eq. 8 inspired by previous works for realization. Namely, the gradient of a neural network w.r.t. the input is orthogonal to the tangent space. According to the well-know property of transformation, we can realize f2 using a composite function, where the utilized transformation models the transformation along local data manifold. Thus, we derive the orthogonal property in Eq. 11.
>
> Most transformations used in the literature can be differentiable by introducing the bilinear interpolation [1,2], usually implemented by the torch.nn.functional.grid_sample function.
>
>
> > Q2. Insufficient Validation of Flow-Based Extrusion (F-ConV)
>
> Thank you for your kind suggestion. To provide a stronger comparison, we additionally trained a binary classifier on the DINOv2 embeddings using the same training set. As shown in the table below, the performance of the directly trained binary classifier falls short of that achieved by F-ConV, highlighting the advantage of our manifold-based approach.
>
> |                       | AUROC | AP    |
> |-----------------------|-------|-------|
> | Linear classification | 87.83 | 86.49 |
> | F-ConV                | 93.77 | 93.38 |
>
> Thanks again for the constructive comment. Accordingly, we have added the above results to our revision.
>
> > Q3. Thresholding Details: The threshold that determines whether an image is synthetic is a critical component of the proposed method. While briefly mentioned in Appendix (Page 14, Line 514), more detail/discussion is needed regarding how this threshold is chosen and whether it generalizes.
>
> Thanks for pointing out this potentially confusing description issue. We have added the following explanations to our revision.
> To optimize overall accuracy, we can establish the classification threshold at a value that maximizes the distinction between natural and generated images. However, in real-world applications, this approach may differ.
>
> For instance, to prioritize the detection of generated images, administrators may set a higher threshold, classifying only samples with similarity scores approaching 1 as natural images. This is consistent with the motivation of leveraging AUROC and AP as the main metric for evaluation.
>
> Regarding the threshold of accuracy, we used the training set from GenImage to select a threshold and applied that threshold to all test sets. As shown in the table below, this approach resulted in a slight performance difference.
>
> |                          | ACC   |
> |--------------------------|-------|
> | ConV varied threshold   | 80.1  |
> | F-ConV varied threshold | 90.1  |
> | ConV fixed threshold     | 79.12 |
> | F-ConV fixed threshold   | 89.43 |
>
>
> > Q4. Performance Variability Across Generative Models: As shown in Tables 1 and 5, the proposed ConV/F-ConV do not consistently outperform all baselines. For images generated by models such as DiT, Stable Diffusion v1.4/1.5, and GLIDE, performance lags behind other methods by a notable margin. A deeper analysis of these failure cases could yield valuable insights into the limitations of the approach and its sensitivity to different generative model architectures or styles.
>
> We speculate that this phenomenon can be intuitively explained using the FID (Fréchet Inception Distance) score. The FID score is a commonly used evaluation metric for generative models, primarily assessing the quality of the generative model by comparing the distribution differences between generated images and natural images in the feature space. A lower FID score indicates that the distribution of the generated images is closer to that of the natural images. As shown in the table below, we have listed the FID scores of several generative models, along with the detection performance of ConV on these models. It is clearly evident that as the FID score of the generative model increases, the detection performance of ConV improves. This is because ConV leverages the differences in data distribution between natural images and generated images for detection. When the differences in data distribution between generated images and natural images are small, directly using ConV may not yield satisfactory results. In such cases, we can employ the proposed F-ConV, which utilizes manifold learning to push generated images out of the natural distribution, thereby increasing the distribution differences between the two.
>
> |          | AUROC | AP    | FID   |
> |----------|-------|-------|-------|
> | ADM      | 88.89 | 86.60 | 11.84 |
> | ADMG     | 82.46 | 79.83 | 5.58  |
> | LDM      | 78.94 | 75.88 | 4.29  |
> | DiT-XL-2 | 75.25 | 70.11 | 2.80  |
>
>
> > Q5. Regarding Figure 2: How are the embeddings generated; What do the “natural images” and “synthetic images” refer to in this visualization?
>
> To visualize the feature distributions of natural and generated images, we use the DINOv2 model to extract their features/embeddings and apply t-SNE for dimensionality reduction. In the resulting visualization, red dots denote features of natural images, while purple dots represent those of generated images.
>
> > Q6. Can the proposed method be extended to detect AI-edited images (e.g., real images with minor synthetic modifications)?
>
>
> Thanks for your inspiring question. We further evaluated the performance of ConV using tampered images from the SD_inpaint dataset. This dataset was constructed by first obscuring regions of natural images and subsequently restoring them using a DDPM-based inpainting model. As shown in the table below, ConV demonstrates strong detection performance on this form of partial image manipulation, highlighting its effectiveness beyond fully generated content.
>
> |      | AUROC | AP    |
> |------|-------|-------|
> | ConV | 83.32 | 84.43 |
>
> > Q7. Page 4, Line 134: The text states: “we denote $x_M(x_g)$ as $x_M$ for simplicity.” Should here be referring to the natural image  $x_M(x_n)$ instead?
>
> We apologize for any confusion. The notation $\mathbf{x}\_{\mathcal{M}}(\mathbf{x}\_g)$ denotes the projection of a generated image $\mathbf{x}\_g$ onto the natural image manifold $\mathcal{M}$; that is, it represents the closest point to $\mathbf{x}\_g$ lying on $\mathcal{M}$. In contrast, a natural image $\mathbf{x}\_n$ is already assumed to reside on the manifold, and thus its projection satisfies $\mathbf{x}\_{\mathcal{M}}(\mathbf{x}\_n) = \mathbf{x}\_n$.
>
>
> ***Reference***
>
> [1] Spatial Transformer Networks, Max Jaderberg, Karen Simonyan, Andrew Zisserman, Koray Kavukcuoglu, NIPS 2015
>
> [2] Deformable Convolutional Networks, Jifeng Dai, Haozhi Qi, Yuwen Xiong, Yi Li, Guodong Zhang, Han Hu, ICCV 2017

---

> ### Author Response · Authors · 2025-08-06
> **Official Comment by Authors**
>
> Dear Reviewer #QsoV,
>
> Thank you very much for your valuable time invested in reviewing our manuscript and for your insightful comments, which have greatly guided our revisions.
>
> We have carefully addressed each of your points in the revised manuscript and the accompanying response, and we hope these modifications adequately address your concerns.
>
> We understand your schedule is extremely busy, but as the discussion deadline is fast approaching, we would be most grateful if you could kindly take a moment to review our response. This would help clarify whether there are any remaining points you wish to discuss further, ensuring we can address them promptly.
>
> Please do not hesitate to let us know if you have any additional questions or require further clarification—we are fully prepared to provide detailed responses at your convenience.
>
> Thank you again for your guidance and support.
>
> Best regards,
>
> The Authors of Paper #16115

---

> > ### Comment · Reviewer_QsoV · 2025-08-06
> >
> > Thank you to the authors for the detailed response and the additional clarifications.
> >
> > 1. **Gap Between Motivation and Implementation**:
> >     I still maintain my concern regarding the gap between the theoretical motivation and the practical implementation. To clarify my point: the implementation appears to rely heavily on the augmentation invariance learned during large-scale self-supervised training on natural images. Specifically, the encoder is encouraged to produce similar outputs for both original and augmented versions of an image. This property, however, is less likely to hold for synthetic images that were unseen during pretraining. In this light, I find the implementation more aligned with the idea of augmentation robustness, rather than the geometric manifold motivation emphasized in the early sections. This disconnect somewhat weakens the relevance of the theoretical motivation. Reviewer 3bZe also raised a related concern in Q1 regarding the lack of empirical validation for the assumption of orthogonal gradient subspaces.
> >
> >    Regarding the differentiability of $f_2$, it would be helpful if the authors could provide a more detailed justification. The current implementation that involves a combination of `HorizontalFlip`, `ColorJitter`, and `GaussianBlur`, is not clearly differentiable, even when "*bilinear interpolation is introduced*". A more explicit explanation would strengthen the validity of the theoretical claims. Along these lines, I am also curious whether the authors considered other commonly used augmentation operators in self-supervised learning, such as `RandomCrop`, `Solarization`, or `Grayscale`, and how they would fit into the framework.
> >
> > 2. **Clarification on Visualization (Q5)**:
> >    Apologies for the lack of clarity in my previous question. I was asking about which set of images were used for feature extraction when generating Figure 2.

---

> > > ### Author Response · Authors · 2025-08-07
> > > **Official Comment by Authors**
> > >
> > > Dear Reviewer QsoV,
> > >
> > > Thank you for your reply. Please find our responses below.
> > >
> > > > I find the implementation more aligned with the idea of augmentation robustness, rather than the geometric manifold motivation emphasized in the early sections.
> > >
> > > Thank you for this constructive observation, which has prompted us to explore deeper insights and empirical evidence supporting our work.
> > >
> > > We agree that new methods can often be interpreted through multiple lenses, and we value such diverse perspectives. Inspired by your insight, we designed an experiment to verify the hypothesis that "_the implementation is more aligned with augmentation robustness than geometric manifold principles._" The following results and analysis will be included in our revised manuscript.
> > >
> > >
> > > While our method shares technical similarities with augmentation robustness, it is fundamentally distinct at a conceptual level. To clarify this, we conducted an experiment using a classification model (ResNet-50 is robust to augmentations but trained to fit semantic information of 1000 classes) alongside models trained to fit natural distributions (MoCo, SwAV, DINOv2) for generated image detection. Our core motivation is fitting natural distributions, so we hypothesized models aligned with this goal would outperform ResNet-50. Results are as follows:
> > >
> > >
> > > |          | AUROC | AP    |
> > > |----------|-------|-------|
> > > | MoCo     | 68.43 | 67.65 |
> > > | SwAV     | 74.71 | 73.48 |
> > > | ResNet50 | 50.46 | 50.29 |
> > > | DINOv2   | 87.13 | 85.15 |
> > >
> > > These results align with our analysis: while augmentation robustness offers an intuitive explanation, it does not capture the core design principles guiding the development of our two functions.
> > >
> > >
> > >
> > > > Reviewer 3bZe also raised a related concern in Q1 regarding the lack of empirical validation for the assumption of orthogonal gradient subspaces.
> > >
> > > Thank you for highlighting this. We have empirically validated the orthogonal gradient subspace assumption: we computed gradients of both $f_1$ and $f_2$ for the same input, then measured their cosine similarity. Across 100 random samples, similarities ranged from 0.0005 to 0.017—consistently low.
> > >
> > > As noted by Reviewer 3bZe, these findings strongly support our theoretical framework and will be included in the revision.
> > >
> > > > Regarding the differentiability of $f_2$, it would be helpful if the authors could provide a more detailed justification.
> > >
> > > $f_2$ achieves differentiability via bilinear interpolation: each pixel in the transformed image is computed as a weighted combination of nearby pixels from the original image, forming a computational graph. This enables end-to-end differentiation.
> > >
> > > We will expand on this in the revised manuscript, and appreciate the reminder.
> > >
> > > > Along these lines, I am also curious whether the authors considered other commonly used augmentation operators in self-supervised learning, such as RandomCrop, Solarization, or Grayscale, and how they would fit into the framework.
> > >
> > > Thank you for this insightful question. We incorporated these three augmentations into our pipeline; preliminary results (below) show they do not improve ConV performance. We hypothesize this is because they introduce significant distortions that disrupt the natural image statistics our framework relies on.
> > >
> > >
> > > |                        | AUROC | AP    |
> > > |------------------------|-------|-------|
> > > | ConV                   | 87.13 | 85.15 |
> > > | ConV + RandomSolarize  | 81.87 | 80.14 |
> > > | ConV + RandomGrayscale | 83.49 | 82.47 |
> > > | ConV + RandomCrop      | 86.47 | 84.65 |
> > >
> > >
> > > > I was asking about which set of images were used for feature extraction when generating Figure 2.
> > >
> > > Thank you for clarifying. For Figure 2, we extracted features from images of birds (to control for category-specific effects) from different generative models in the GenImage training set.
> > >
> > > We appreciate your detailed feedback and will incorporate all these clarifications and results into the revised manuscript.
> > >
> > > Best regards,
> > >
> > > The Authors of Paper #16115

---

> > > > ### Comment · Reviewer_QsoV · 2025-08-07
> > > >
> > > > Thank you for the prompt and detailed follow-up, and for running the additional experiments.
> > > >
> > > > 1. **Interpretation of the SSL vs. supervised comparison**
> > > >
> > > >    The new results are helpful, yet I believe they can still be explained from the viewpoint of augmentation invariance. Self-supervised objectives (MoCo, SwAV, DINOv2) explicitly encourage the encoder to become invariant to their own augmentations, which closely aligns with the practical realization of F-ConV. In contrast, a supervised ResNet-50 is trained to discriminate 1000 ImageNet categories and therefore may learn coarser, class-level invariances. This alternative interpretation does not necessarily contradict your motivation, but suggests that augmentation robustness remains a simpler explanation of the observed behavior.
> > > >
> > > > 2. **Differentiability of $f_2$ and gradient-orthogonality verification**
> > > >
> > > >    Could the author please elaborate on the exact procedure used to compute the gradients of $f_1$​ and $f_2$​? Am I correct that the gradients you report are $\nabla_{\mathbf{x}} f_i(\mathbf{x})$, evaluated with respect to the unaugmented input tensor and averaged over the network parameters?  Introducing bilinear interpolation could makes continuous spatial transforms (e.g., small translations or scaling) differentiable, but it is not obvious for discrete operations such as horizontal flips or certain color manipulations. Do you re-implement these operators with differentiable kernels, or do you approximate them in another way? A brief description (or pseudocode) of how the augmentation in the pipeline is made differentiable would be very helpful.
> > > >
> > > >
> > > >
> > > > Thank you again for the careful revisions.

---

> > > > > ### Author Response · Authors · 2025-08-08
> > > > > **Official Comment by Authors**
> > > > >
> > > > > ```python
> > > > > import torch
> > > > > import torch.nn as nn
> > > > > import torch.nn.functional as F
> > > > > import math
> > > > >
> > > > > _RGB2YUV = torch.tensor([[ 0.299,    0.587,    0.114 ],
> > > > >                          [-0.14713, -0.28886,  0.436 ],
> > > > >                          [ 0.615,   -0.51499, -0.10001]], dtype=torch.float32)
> > > > > _YUV2RGB = torch.tensor([[1.0,   0.0,      1.13983],
> > > > >                          [1.0,  -0.39465, -0.58060],
> > > > >                          [1.0,   2.03211,  0.0    ]], dtype=torch.float32)
> > > > >
> > > > > class DifferentiableColorTransform(nn.Module):
> > > > >     def __init__(self,
> > > > >                  brightness_range=(0.0, 0.2),
> > > > >                  contrast_range=(0.8, 1.2),
> > > > >                  saturation_range=(0.8, 1.2),
> > > > >                  hue_range=(-0.1, 0.1)):
> > > > >         super().__init__()
> > > > >         self.brightness_range = brightness_range
> > > > >         self.contrast_range   = contrast_range
> > > > >         self.saturation_range = saturation_range
> > > > >         self.hue_range        = hue_range
> > > > >
> > > > >         self.register_buffer('rgb2yuv', _RGB2YUV.clone())
> > > > >         self.register_buffer('yuv2rgb', _YUV2RGB.clone())
> > > > >         self.register_buffer('luma', torch.tensor([0.299, 0.587, 0.114]).view(1,3,1,1))
> > > > >
> > > > >     def forward(self, x: torch.Tensor) -> torch.Tensor:
> > > > >
> > > > >         B, C, H, W = x.shape
> > > > >         device, dtype = x.device, x.dtype
> > > > >
> > > > >         luma    = self.luma.to(device=device, dtype=dtype)
> > > > >         rgb2yuv = self.rgb2yuv.to(device=device, dtype=dtype)
> > > > >         yuv2rgb = self.yuv2rgb.to(device=device, dtype=dtype)
> > > > >
> > > > >
> > > > >         def ru(lo, hi, shape):
> > > > >             return lo + (hi - lo) * torch.rand(shape, device=device, dtype=dtype)
> > > > >
> > > > >         b     = ru(self.brightness_range[0], self.brightness_range[1], (B,1,1,1))
> > > > >         c     = ru(self.contrast_range[0],   self.contrast_range[1],   (B,1,1,1))
> > > > >         s     = ru(self.saturation_range[0], self.saturation_range[1], (B,1,1,1))
> > > > >         theta = ru(self.hue_range[0],        self.hue_range[1],        (B,1,1,1))
> > > > >
> > > > >
> > > > >         mu = (x * luma).sum(dim=1, keepdim=True)
> > > > >         x  = (x - mu) * c + mu + b
> > > > >
> > > > >
> > > > >         gray = (x * luma).sum(dim=1, keepdim=True)
> > > > >         x    = gray + s * (x - gray)
> > > > >
> > > > >
> > > > >         x_perm = x.permute(0,2,3,1)
> > > > >         YUV    = torch.matmul(x_perm, rgb2yuv.T)
> > > > >         Y = YUV[..., 0]
> > > > >         U = YUV[..., 1]
> > > > >         V = YUV[..., 2]
> > > > >
> > > > >
> > > > >         cos_t = torch.cos(theta).view(B,1,1).expand(B,H,W)
> > > > >         sin_t = torch.sin(theta).view(B,1,1).expand(B,H,W)
> > > > >
> > > > >         U_rot = cos_t * U - sin_t * V
> > > > >         V_rot = sin_t * U + cos_t * V
> > > > >
> > > > >         YUV_r = torch.stack([Y, U_rot, V_rot], dim=-1)
> > > > >         RGB   = torch.matmul(YUV_r, yuv2rgb.T)
> > > > >         x     = RGB.permute(0,3,1,2)
> > > > >         return x
> > > > >
> > > > >
> > > > >
> > > > >
> > > > > class DifferentiableFlipTransform(nn.Module):
> > > > >     def __init__(self):
> > > > >         super().__init__()
> > > > >
> > > > >     def horizontal_flip(self, image):
> > > > >
> > > > >         B, C, H, W = image.shape
> > > > >
> > > > >         grid_y, grid_x = torch.meshgrid(
> > > > >             torch.linspace(-1, 1, H, device=image.device),
> > > > >             torch.linspace(-1, 1, W, device=image.device),
> > > > >             indexing='ij'
> > > > >         )
> > > > >
> > > > >         grid = torch.stack((-grid_x, grid_y), dim=-1)
> > > > >         grid = grid.unsqueeze(0).repeat(B, 1, 1, 1)
> > > > >         flipped = torch.nn.functional.grid_sample(image, grid, mode='bilinear', align_corners=False)
> > > > >         return flipped
> > > > >
> > > > >     def vertical_flip(self, image):
> > > > >
> > > > >         B, C, H, W = image.shape
> > > > >
> > > > >         grid_y, grid_x = torch.meshgrid(
> > > > >             torch.linspace(-1, 1, H, device=image.device),
> > > > >             torch.linspace(-1, 1, W, device=image.device),
> > > > >             indexing='ij'
> > > > >         )
> > > > >
> > > > >
> > > > >         grid = torch.stack((grid_x, -grid_y), dim=-1)
> > > > >         grid = grid.unsqueeze(0).repeat(B, 1, 1, 1)
> > > > >
> > > > >         flipped = torch.nn.functional.grid_sample(image, grid, mode='bilinear', align_corners=False)
> > > > >         return flipped
> > > > >
> > > > >     def forward(self, x, flip_type='horizontal'):
> > > > >
> > > > >         if flip_type == 'horizontal':
> > > > >             return self.horizontal_flip(x)
> > > > >         elif flip_type == 'vertical':
> > > > >             return self.vertical_flip(x)
> > > > >         elif flip_type == 'both':
> > > > >             x_h = self.horizontal_flip(x)
> > > > >             return self.vertical_flip(x_h)
> > > > >         else:
> > > > >             return x

---

> > > > > ### Author Response · Authors · 2025-08-08
> > > > > **Official Comment by Authors**
> > > > >
> > > > > ```python
> > > > > class DifferentiableAffineTransform(nn.Module):
> > > > >     def __init__(self,
> > > > >                  rotation_range=(-10.0, 10.0),
> > > > >                  scale_range=(0.95, 1.05),
> > > > >                  translation_range=(-0.05, 0.05),
> > > > >                  shear_x_range=(-5.0, 5.0),
> > > > >                  shear_y_range=(-5.0, 5.0),
> > > > >                  padding_mode='border'):
> > > > >         super().__init__()
> > > > >         self.rotation_range    = rotation_range
> > > > >         self.scale_range       = scale_range
> > > > >         self.translation_range = translation_range
> > > > >         self.shear_x_range     = shear_x_range
> > > > >         self.shear_y_range     = shear_y_range
> > > > >         self.padding_mode      = padding_mode
> > > > >
> > > > >     @staticmethod
> > > > >     def _ru(lo, hi, shape, device, dtype):
> > > > >         return lo + (hi - lo) * torch.rand(shape, device=device, dtype=dtype)
> > > > >
> > > > >     def forward(self, x: torch.Tensor) -> torch.Tensor:
> > > > >         B, C, H, W = x.shape
> > > > >         device, dtype = x.device, x.dtype
> > > > >
> > > > >
> > > > >         ang_deg = self._ru(*self.rotation_range,    (B,), device, dtype)
> > > > >         sx      = self._ru(*self.scale_range,       (B,), device, dtype)
> > > > >         sy      = self._ru(*self.scale_range,       (B,), device, dtype)
> > > > >         tx      = self._ru(*self.translation_range, (B,), device, dtype)
> > > > >         ty      = self._ru(*self.translation_range, (B,), device, dtype)
> > > > >         shx_deg = self._ru(*self.shear_x_range,     (B,), device, dtype)
> > > > >         shy_deg = self._ru(*self.shear_y_range,     (B,), device, dtype)
> > > > >
> > > > >         ang = ang_deg * (math.pi/180.0)
> > > > >         shx = shx_deg * (math.pi/180.0)
> > > > >         shy = shy_deg * (math.pi/180.0)
> > > > >
> > > > >         cos_a, sin_a = torch.cos(ang), torch.sin(ang)
> > > > >         tan_shx, tan_shy = torch.tan(shx), torch.tan(shy)
> > > > >
> > > > >
> > > > >         cx = (W - 1) / 2.0
> > > > >         cy = (H - 1) / 2.0
> > > > >
> > > > >         sxn = 2.0 / (W - 1)
> > > > >         syn = 2.0 / (H - 1)
> > > > >
> > > > >         cx_n = (cx * sxn)
> > > > >         cy_n = (cy * syn)
> > > > >
> > > > >         I = torch.eye(3, device=device, dtype=dtype).unsqueeze(0).expand(B,3,3).clone()
> > > > >
> > > > >
> > > > >         T_c   = I.clone()
> > > > >         T_c[:,0,2] = cx_n
> > > > >         T_c[:,1,2] = cy_n
> > > > >
> > > > >         T_ci  = I.clone()
> > > > >         T_ci[:,0,2] = -cx_n
> > > > >         T_ci[:,1,2] = -cy_n
> > > > >
> > > > >
> > > > >         S = I.clone()
> > > > >         S[:,0,0] = sx
> > > > >         S[:,1,1] = sy
> > > > >
> > > > >
> > > > >         R = I.clone()
> > > > >         R[:,0,0] =  cos_a
> > > > >         R[:,0,1] = -sin_a
> > > > >         R[:,1,0] =  sin_a
> > > > >         R[:,1,1] =  cos_a
> > > > >
> > > > >
> > > > >         Shx = I.clone()
> > > > >         Shx[:,0,1] = tan_shx
> > > > >         Shy = I.clone()
> > > > >         Shy[:,1,0] = tan_shy
> > > > >         Sh = Shy @ Shx
> > > > >
> > > > >
> > > > >         T = I.clone()
> > > > >         T[:,0,2] = tx
> > > > >         T[:,1,2] = ty
> > > > >
> > > > >
> > > > >         M = T @ T_c @ Sh @ R @ S @ T_ci
> > > > >
> > > > >
> > > > >         theta = M[:, :2, :]
> > > > >
> > > > >         grid = F.affine_grid(theta, size=x.size(), align_corners=False)
> > > > >
> > > > >
> > > > >         y = F.grid_sample(x, grid, mode='bilinear',
> > > > >                           padding_mode=self.padding_mode, align_corners=False)
> > > > >         return y
> > > > > ```
> > > > >
> > > > > Best regards,
> > > > >
> > > > > The Authors of Paper #16115

---

> > > > > > ### Comment · Reviewer_QsoV · 2025-08-08
> > > > > >
> > > > > > Thank you to the authors for their prompt response and for sharing the implementation details. While I still maintain my view that, based on the current implementation, augmentation invariance for natural images in SSL models could serve as an equally appropriate and perhaps more intuitive theoretical foundation (in the sense that one could develop a paper with a similar implementation/experiment but framed under this alternative motivation, and potentially making the theoretical justification more straightforward), most of my concerns regarding the proposed motivation have been addressed. I have accordingly adjusted my rating upward. I also appreciate the authors’ timely and thorough engagement throughout our discussion.

---

> > > > > > > ### Author Response · Authors · 2025-08-09
> > > > > > > **Official Comment by Authors**
> > > > > > >
> > > > > > > Dear Reviewer QsoV,
> > > > > > >
> > > > > > > Thank you sincerely for your thoughtful reply and for taking the time to adjust your rating.
> > > > > > >
> > > > > > > Your insights throughout this discussion have been invaluable—your deep engagement with our work, incisive observations, and the alternative perspective on augmentation invariance in SSL models have significantly elevated the quality of our work.
> > > > > > >
> > > > > > > Thank you again for your timely, thorough, and constructive engagement. Your guidance has been instrumental in refining our work, and we will ensure to incorporate these insights thoughtfully in the revised manuscript.
> > > > > > >
> > > > > > > Best regards,
> > > > > > >
> > > > > > > The Authors of Paper #16115

---

> ### Author Response · Authors · 2025-08-08
> **Official Comment by Authors**
>
> Dear Reviewer QsoV,
>
> Thank you for your thoughtful engagement and the opportunity for this valuable technical discussion.
>
> > **Addressing Concern 1**: _The new results are helpful, yet I believe they can still be explained from the viewpoint of augmentation invariance. This alternative interpretation does not necessarily contradict your motivation, but suggests that augmentation robustness remains a simpler explanation of the observed behavior._
>
> We appreciate your perspective connecting our approach to augmentation robustness in self-supervised learning. This insight aligns with prior work [1,2], and we agree that our method can be intuitively understood through the lens of augmentation invariance, grounding it in these established principles [1,2].
>
>
> > **Addressing Concern 2**: _Could the author please elaborate on the exact procedure used to compute the gradients of $f_1$​ and $f_2$​? Am I correct that the gradients you report are $\nabla\_{\mathbf{x}} f_i(\mathbf{x})$, evaluated with respect to the unaugmented input tensor and averaged over the network parameters?_
>
> Thank you for this important clarification request. In our method, gradients are computed with respect to the original input pixels (not model parameters), yielding gradient maps with the same spatial dimensions as the input image. We then measure the cosine similarity between these two gradient maps.
>
>
> > **Implementation of Differentiable Transformations**: _It is not obvious for discrete operations such as horizontal flips or certain color manipulations. A brief description (or pseudocode) of how the augmentation in the pipeline is made differentiable would be very helpful._
>
> Following your valuable suggestion, we provide implementation details for our differentiable transformation pipeline:
>
> - Differentiable affine transforms: rotation, scaling, translation, and shearing;
>
> - Differentiable color transforms: brightness, contrast, saturation, and hue adjustments;
>
> - Differentiable flip transforms: horizontal and vertical flips.
>
> All transformations maintain end-to-end differentiability through PyTorch's automatic differentiation, enabling gradient computation through the entire pipeline.
>
> The connection to augmentation robustness you identified offers novel theoretical and empirical perspectives for our proposed orthogonal principle, paired with its realization using data augmentation.
>
> We hope these clarifications address your concerns and highlight our contributions to the field. We would greatly appreciate your consideration of these details in your final assessment.
>
> [1] Provable Guarantees for Self-Supervised Deep Learning with Spectral Contrastive Loss, NeurIPS 2021
>
> [2] Understanding Augmentation-based Self-Supervised Representation Learning via RKHS Approximation and Regression, ICLR 2024

---

### Official Review · Reviewer_xMhJ · 2025-07-02

**Clarity:** 2
**Significance:** 3
**Originality:** 3
**Rating:** 4
**Confidence:** 4

**Summary:**

The core concept of **ConV** involves analyzing the data manifolds of natural images instead of depending on data from generated images, thereby avoiding the reliance of current methods on generated image datasets. This improves the generalizability of the detection methods, particularly when dealing with unknown generative models. The proposed approach leverages pre-trained self-supervised models (e.g., DINOv2) to extract features from images and assesses whether an image is synthesized by measuring the feature consistency between the original image and its transformed variants. Being training-free, it reduces computational costs and enhances flexibility in deployment. Also, the using of Normalizing Flows further enhances detection performance.

**Questions:**

1. Does this method remain effective when applied to larger and more recent models, such as Flux, that are trained on extensive datasets like LAION?
2. Why does the performance of the training-free method on DiT significantly lag behind that of the baseline in Tab.1?

**Ethical Concerns:**

["NO or VERY MINOR ethics concerns only"]

**Final Justification:**

Most of my concerns have been well addressed, and as a result, I have increased my score.

**Limitations:**

No limitations discussed in this paper.

**Quality:**

2

**Strengths And Weaknesses:**

Strengths:
1. Generalization Capability: ConV is independent of specific generative models, enabling it to excel when confronted with unknown generative models. In particular, it has demonstrated excellent performance in experiments detecting unpublished models such as Sora.
2. Computational Efficiency: The training-free nature of this method results in high efficiency during practical applications, making it ideal for environments with limited computational resources.

Weaknesses:
1. The writing in this paper requires significant improvement.  Some sentences are confusing and difficult to read, such as the first paragraph of Section 2.1. Also, there is missing information; for instance, in Figure 2, it is unclear which color represents real images and which represents fake images. There are also formatting issues, such as inconsistent spacing observed in line 130. Moreover, there are inconsistencies in notation usage. For example, in line 107, "x_n" is used to denote natural images, but throughout the rest of the paper, "x_m" appears to be used instead.
2. Although ConV achieves strong results as indicated in Table 1, it seems that the primary contributing factor is the pre-trained model, such as DINOv2 (refer to line 269). This factor should not be disregarded and warrants further discussion. This is my major concern about this paper.
3. Why is the first item, starting on line 81, considered the main contribution of this paper?

---

> ### Author Rebuttal · Authors · 2025-07-31
>
> We are truly grateful for the reviewers' dedicated efforts. The valuable comments greatly improve the quality of our work. We provide the following answers to the reviewers' concerns.
>
>
> > Q1. The writing in this paper requires significant improvement. Some sentences are confusing and difficult to read, such as the first paragraph of Section 2.1. Also, there is missing information; for instance, in Figure 2, it is unclear which color represents real images and which represents fake images. There are also formatting issues, such as inconsistent spacing observed in line 130. Moreover, there are inconsistencies in notation usage. For example, in line 107, "x_n" is used to denote natural images, but throughout the rest of the paper, "x_m" appears to be used instead.
>
> We express our gratitude for the reviewer’s kind suggestion. We have added the following clarification to our revision.
> - In Figure 2, red dots denote the feature representations of natural images, while purple dots represent those of generated images.
> - The terms $\mathbf{x}\_{\mathcal{M}}$ and $\mathbf{x}\_{\mathcal{n}}$ convey distinct meanings: $\mathbf{x}\_{\mathcal{n}}$ corresponds to the feature representation of a natural image, whereas $\mathbf{x}\_{\mathcal{M}}$ represents the projection of a generated image’s features onto the manifold of natural images.
>
> > Q2. Although ConV achieves strong results as indicated in Table 1, it seems that the primary contributing factor is the pre-trained model, such as DINOv2 (refer to line 269). This factor should not be disregarded and warrants further discussion. This is my major concern about this paper.
>
> Thank you for your insightful comment. To further validate the effectiveness of our proposed method, we conducted additional experiments across a range of backbone models, including MoCo, SwAV, DINO, CLIP, and SigLIP.
>
> ConV operates without any additional training and relies solely on a pre-trained feature extractor. Its performance is therefore contingent on the extractor’s ability to strike a delicate balance: accurately capturing the intrinsic structure of natural images while avoiding overgeneralization to generated content. As a result, ConV is inherently sensitive to the choice of feature extractor, as demonstrated in the results reported below.
>
> |                             | AUROC | AP    |
> |-----------------------------|-------|-------|
> | MoCo                      | 68.43 | 67.65 |
> | SwAV     |   74.71   |  73.48     |
> | DINO |    71.91   |  69.46     |
> | ConV DINOv2 ViT-L/14   | 87.13 | 85.15 |
> | ConV CLIP ViT-L/14     |  72.99  | 72.98      |
> | ConV DionV2-vit-b      |   78.58    | 76.39      |
> |SigLIP|84.19|81.93|
>
> Nevertheless, this sensitivity can be substantially alleviated through fine-tuning the feature extractor, as further evidenced by the performance improvements shown in the accompanying table.
>
>
> |                        | AUROC | AP    |
> |------------------------|-------|-------|
> | ConV DINOv2 ViT-L/14   | 87.13 | 85.15 |
> | ConV CLIP ViT-L/14     |  72.99  | 72.98      |
> | ConV DionV2-vit-b      |   78.58    | 76.39      |
> | F-ConV DINOv2 ViT-L/14 | 93.77 | 93.38 |
> | F-ConV CLIP ViT-L/14   |   94.25  |  94.11   |
> | F-ConV DINOv2 ViT-B/14 |   89.17    |  88.54     |
>
> Thanks again for your constructive comments. Accordingly, the above results have been added to our revision.
>
> > Q3. Why is the first item, starting on line 81, considered the main contribution of this paper?
>
> Most AI-generated image detection methods involve training a binary classifier for natural images and generated images. This approach implicitly assumes that generated images share certain shared features. However, as generative models continue to evolve, this assumption may be challenged. Instead, in this paper, we focus on how to distinguish between natural images and generated images by simply fitting the natural distribution, thereby avoiding any prior assumptions about generated images.
>
>
> > Q4. Does this method remain effective when applied to larger and more recent models, such as Flux, that are trained on extensive datasets like LAION?
>
> Thank you for your constructive suggestion. We further assessed the performance of ConV on the Flux model by constructing a new evaluation dataset comprising 6,000 images generated by the FLUX.1 [dev] model, alongside natural images sampled from ImageNet. As shown in the table below, ConV demonstrates strong performance on this previously unseen generative model, further highlighting the generalizability of our approach.
>
> |        | AUROC | AP    |
> |--------|-------|-------|
> | ConV   | 87.38 | 89.19 |
> | F-ConV | 90.18 | 90.65   |
>
> > Q5. Why does the performance of the training-free method on DiT significantly lag behind that of the baseline in Tab.1?
>
> We speculate that this phenomenon can be intuitively explained using the FID (Fréchet Inception Distance) score. The FID score is a commonly used evaluation metric for generative models, primarily assessing the quality of the generative model by comparing the distribution differences between generated images and natural images in the feature space. A lower FID score indicates that the distribution of the generated images is closer to that of the natural images. As shown in the table below, we have listed the FID scores of several generative models, along with the detection performance of ConV on these models. It is clearly evident that as the FID score of the generative model increases, the detection performance of ConV improves. This is because ConV leverages the differences in data distribution between natural images and generated images for detection. When the differences in data distribution between generated images and natural images are small, directly using ConV may not yield satisfactory results. In such cases, we can employ the proposed F-ConV, which utilizes manifold learning to push generated images out of the natural distribution, thereby increasing the distribution differences between the two.
>
> |          | AUROC | AP    | FID   |
> |----------|-------|-------|-------|
> | ADM      | 88.89 | 86.60 | 11.84 |
> | ADMG     | 82.46 | 79.83 | 5.58  |
> | LDM      | 78.94 | 75.88 | 4.29  |
> | DiT-XL-2 | 75.25 | 70.11 | 2.80  |

---

> > ### Comment · Reviewer_xMhJ · 2025-08-05
> >
> > Thank you for the author's response. Most of my concerns have been well addressed, and as a result, I have increased my score.
> > However, I still somewhat disagree with the answer to Q3, where the author stated, `as generative models continue to evolve, this assumption may be challenged.` I believe that one of the primary reasons generative models continue to advance is the ongoing improvement of natural images. Hence, the natural images are also changing.

---

> > > ### Author Response · Authors · 2025-08-05
> > >
> > > Thank you for your prompt reply. We are pleased to have addressed your concerns and sincerely appreciate your willingness to consider increasing the score. We have carefully incorporated your feedback into the revised manuscript, which we believe has significantly enhanced the clarity, rigor, and overall quality of the paper.
> > >
> > > With regard to your observation about the continuous evolution of natural images, we find it both insightful and highly informative. We share your view that natural images will play an increasingly significant role in shaping future research in this field, and we remain optimistic about the continued growth and success of our community.

---

### Official Review · Reviewer_5UgF · 2025-07-02

**Clarity:** 2
**Significance:** 3
**Originality:** 3
**Rating:** 4
**Confidence:** 3

**Summary:**

This paper proposes a method for detecting generated images. It proposes two versions: ConV, a fully training-free approach that compares the feature similarity of an image and its transformed versions. If they are dissimilar, an image is likely generated. Another version is F-ConV, which supplements ConV by training a normalizing flow model that separates the natural and generated image manifolds, and the decision is made by additionally considering the likelihood under this model. Experiments compare ConV and F-ConV with a variety of baselines on various datasets and scenarios, and results show that the proposed methods attain consistent and strong performance.

**Questions:**

Decision threshold selection in practice:
* L542 mentions that the decision threshold is selected automatically to maximize the separation between natural and generated images. Is this performed per-domain/dataset or generally across all datasets/domains?
* Related to this, how does the cosine similarity distribution for natural vs. generated images compare across domains and datasets? Are there cases where similarities for generated images of one dataset are higher than those of natural images of another dataset? If this is the case, in practical, generic situations where natural images encompass all non-generated images, and synthetic images can come from any model with any training data, how should the decision threshold be decided?

Ablation on data transformation:
* While it is intuitive to use the same transformations as those used in the feature extractor’s training, are there any empirical analysis to determine if all of them are truly necessary, or are specific ones that contribute more and would suffice (holding the number of transformations constant)?

**Ethical Concerns:**

["NO or VERY MINOR ethics concerns only"]

**Final Justification:**

Thank the authors for the rebuttal. I am satisfied with the response and decide to keep my score of borderline accept.

**Limitations:**

Yes the authors have discussed potential limitations.

**Paper Formatting Concerns:**

No significant formatting concern.

**Quality:**

3

**Strengths And Weaknesses:**

Strengths:
* The proposed methods are well-motivated, intuitive and straightforward.
* Evaluations are extensive, covering diverse datasets, corruption scenarios and baselines.
* Results show that the proposed methods achieve strong and consistent performance. The training-free version ConV attains comparable performance to various training-based methods, and the training version F-ConV attains competitive performance over existing methods.

Weaknesses:
* The performance is sensitive to the choice of feature extractor. Using smaller DINOv2 models (L269) or other types of feature extractors (supp. Table 8) as the backbone leads to significantly reduced performance.
* Additional clarifications can be made regarding decision threshold selection and ablation on transformations (see Questions below).

---

> ### Author Rebuttal · Authors · 2025-07-31
>
> We are truly grateful for the reviewers' dedicated efforts. The valuable comments greatly improve the quality of our work. We provide the following answers to the reviewers' concerns.
>
>
> > Q1. The performance is sensitive to the choice of feature extractor.
>
> We express our gratitude for the reviewer’s insightful comment regarding the sensitivity of our method to the choice of feature extractor. As ConV operates without additional training and relies solely on a pre-trained feature extractor, its performance hinges on the extractor’s ability to maintain a precise balance: effectively capturing the characteristics of natural images while avoiding overgeneralization to generated images. This results in ConV being sensitive to the feature extractor used. However, this sensitivity can be mitigated through fine-tuning the feature extractor, as evidenced by the results presented in the table below.
>
> |                        | AUROC | AP    |
> |------------------------|-------|-------|
> | ConV DINOv2 ViT-L/14   | 87.13 | 85.15 |
> | ConV CLIP ViT-L/14     |  72.99  | 72.98      |
> | ConV DionV2-vit-b      |   78.58    | 76.39      |
> | F-ConV DINOv2 ViT-L/14 | 93.77 | 93.38 |
> | F-ConV CLIP ViT-L/14   |   94.25  |  94.11   |
> | F-ConV DINOv2 ViT-B/14 |   89.17    |  88.54     |
>
> Thanks again for your inspiring comments. We have added the above results to our revision.
>
>
> > Q2. L542 mentions that the decision threshold is selected automatically to maximize the separation between natural and generated images. Is this performed per-domain/dataset or generally across all datasets/domains?
>
> Thanks for pointing out this potentially confusing description issue. We have added the following explanations to our revision.
> To optimize overall accuracy, we can establish the classification threshold at a value that maximizes the distinction between natural and generated images. However, in real-world applications, this approach may differ.
>
> For instance, to prioritize the detection of generated images, administrators may set a higher threshold, classifying only samples with similarity scores approaching 1 as natural images. This is consistent with the motivation of leveraging AUROC and AP as the main metric for evaluation.
>
> Regarding the threshold of accuracy, we used the training set from GenImage to select a threshold and applied that threshold to all test sets. As shown in the table below, this approach resulted in a slight performance difference.
>
> |                          | ACC   |
> |--------------------------|-------|
> | ConV varied threshold   | 80.1  |
> | F-ConV varied threshold | 90.1  |
> | ConV fixed threshold     | 79.12 |
> | F-ConV fixed threshold   | 89.43 |
>
>
>
> > Q3. Related to this, how does the cosine similarity distribution for natural vs. generated images compare across domains and datasets? Are there cases where similarities for generated images of one dataset are higher than those of natural images of another dataset? If this is the case, in practical, generic situations where natural images encompass all non-generated images, and synthetic images can come from any model with any training data, how should the decision threshold be decided?
>
> We acknowledge that generated images can exhibit similarity scores surpassing those of natural images, which precludes achieving a 100% accuracy rate. In our experiments, to optimize overall accuracy, we establish the classification threshold at a value that maximizes the distinction between natural and generated images. However, in real-world applications, this approach may differ. For instance, to prioritize the detection of generated images, administrators may set a higher threshold, classifying only samples with similarity scores approaching 1 as natural images.
>
>
> > Q4. While it is intuitive to use the same transformations as those used in the feature extractor’s training, are there any empirical analysis to determine if all of them are truly necessary, or are specific ones that contribute more and would suffice (holding the number of transformations constant)?
>
> We express our appreciation for the reviewer’s valuable suggestion. To further investigate the impact of data transformations, we conducted additional ablation studies on the transformation functions employed. The experimental results show that the three transformations all have an impact on the detection effect.
>
> |                         | AUROC | AP    |
> |-------------------------|-------|-------|
> | ConV                    | 87.13 | 85.15 |
> | ConV w/o HorizontalFlip | 86.55 | 84.19 |
> | ConV w/o ColorJitter    | 85.97 | 83.74 |
> | Conv w/o GaussianBlur   | 85.49 | 82.83 |

---

> > ### Comment · Reviewer_5UgF · 2025-08-05
> >
> > Thank the authors for the rebuttal. They answered my questions and I am satisfied with the response. Therefore I decide to keep my score of borderline accept.

---

> > > ### Author Response · Authors · 2025-08-05
> > >
> > > We sincerely thank the reviewer for the valuable comments, which have significantly improved the quality of our paper. We also appreciate the reviewer’s acknowledgment of our efforts during the discussion. Should there be any further questions, we would be glad to address them to continue improving our work. Thank you for your time and thoughtful review.

---

### Official Review · Reviewer_3bZe · 2025-07-05

**Clarity:** 3
**Significance:** 3
**Originality:** 4
**Rating:** 5
**Confidence:** 3

**Summary:**

This paper presents ConV, a training-free framework for detecting generated images by exploiting the geometric inconsistency between natural and synthetic image manifolds. The key insight is that natural images maintain feature consistency under small, manifold-preserving transformations, while generated images exhibit notable representation shifts. By comparing features of an image and its augmented version using a pre-trained vision backbone (e.g., DINOv2), ConV effectively distinguishes real from fake content. To further enhance performance, F-ConV extends this framework with a flow-based manifold extrusion mechanism: using normalizing flows, it reshapes the complex natural image distribution into a Gaussian space, enabling explicit separation from generated content. Extensive experiments show that ConV performs on par with or better than many training-based baselines, especially under distribution shifts and on unseen generative models (e.g., Sora). Visualizations with t-SNE and cosine similarity confirm that natural images retain stable representations, while generated images deviate significantly after transformation, validating the core assumption behind ConV.

**Questions:**

1. How sensitive is the method to the choice of the transformation function $h(\cdot)$? Are some augmentations more effective than others?
2. Can the authors provide empirical validation for the assumption that the gradients lie in orthogonal subspaces for natural vs. generated images?
3. How does the detection performance change if a smaller or weaker self-supervised backbone is used instead of DINOv2?
4. Could the authors provide ablation studies isolating the shaping and consistency losses in F-ConV?
5. How is the performance better than a simpler setup where we train an autoencoder over training images and detect generated images if the loss is higher than a threshold?

**Ethical Concerns:**

["NO or VERY MINOR ethics concerns only"]

**Final Justification:**

The authors have addressed all of the reviewer’s concerns. To this end, the reviewer is raising the original rating.

**Limitations:**

yes

**Paper Formatting Concerns:**

No issue with the formatting is spotted.

**Quality:**

4

**Strengths And Weaknesses:**

**Strenghts**
* The use of normalizing flows is well-motivated and novel. The proposed F-ConV framework introduces a principled use of normalizing flows to map natural image features into a Gaussian latent space. This invertible transformation allows generated images to be effectively extruded from the natural image manifold, amplifying their separability. The design is both novel and technically sound, improving the core ConV framework.
* The proposed method demonstrates strong generalization to unseen generative models. ConV shows robustness on evaluations with real-world video frames generated by black-box models, i.e., Sora and OpenSora. These results highlight the method's ability to handle significant distribution shifts and unseen generative sources without requiring re-training or domain adaptation, which is a very important property in real-world scenarios.
* Insightful visualizations supporting intuition. t-SNE visualizations are provided that illustrate feature (in)consistency under data transformations. These plots provide evidence for the underlying mechanism and the core idea of the paper, i.e., that natural images exhibit consistent representations under manifold-preserving transformations, whereas generated images do not. This strengthens the paper's clarity and helps understand the behavior of the method.

**Weaknesses**
1. Assumption of orthogonal gradient subspaces lacks empirical validation. A core idea for the method is that gradients with respect to two designed functions lie in orthogonal subspaces for natural vs. generated images. While this is sound, it is not empirically validated. This leaves a gap between the method's theoretical motivation and empirical evidence.

2. Generalization with other backbones is not assessed. While the method achieves strong results using DINOv2, it remains unclear how sensitive ConV is to the choice of backbone. There is limited investigation into how the method would perform with weaker or less recent self-supervised models. This raises concerns about the generality of the method and the transferability of conclusions to other cases, especially in scenarios where DINOv2 is unavailable or too resource-intensive.

3. F-ConV loss components are not ablated. The F-ConV extension introduces a loss combining manifold shaping and consistency terms. However, no clear ablation study is provided to disentangle the contribution of each component. It is not clear how much each term contributes to the performance gain, limiting insight into the effectiveness and necessity of the loss design.

4. No analysis of failure cases is provided. Although the method shows strong empirical results, no discussion of failure cases is provided. For example, high-quality generated images that preserve feature consistency under transformation, or natural images degraded by noise or low-light conditions, may challenge the method's assumptions. Understanding when and why the method fails would be valuable for improving robustness.

5. The heavy mathematical formalism might limit accessibility. The core ideas, i.e., leveraging manifold geometry, enforcing gradient orthogonality, and detecting inconsistency under transformation, are insightful; however, the presentation is mathematically dense, with technical details involving Jacobians, projections, and flow-based latent space transformations. This level of formalism may make it difficult for the broader audience to grasp the main intuition of the paper.

---

> ### Author Rebuttal · Authors · 2025-07-31
>
> We are truly grateful for the reviewers' dedicated efforts. The valuable comments greatly improve the quality of our work. We provide the following answers to the reviewers' concerns.
>
>
> > Q1. Assumption of orthogonal gradient subspaces lacks empirical validation. Can the authors provide empirical validation for the assumption that the gradients lie in orthogonal subspaces for natural vs. generated images?
>
> Thanks for your constructive feedback! The mentioned results would provide a straightforward approach to verify the effectiveness of our method, thereby significantly improving the quality of our work. We will provide the complete empirical validation results during the discussion period.
>
>
> > Q2. Generalization with other backbones is not assessed. How does the detection performance change if a smaller or weaker self-supervised backbone is used instead of DINOv2?
>
>
> We express our gratitude for the reviewer’s insightful comment. To further substantiate the efficacy of our proposed method, we conducted additional experiments using various backbone models, including MoCo, SwAV, DINO, CLIP, and SigLIP. The experimental results, presented in the table below, demonstrate that ConV maintains robust detection performance even when applied to less powerful models.
>
> |                             | AUROC | AP    |
> |-----------------------------|-------|-------|
> | MoCo                      | 68.43 | 67.65 |
> | SwAV     |   74.71   |  73.48     |
> | DINO |    71.91   |  69.46     |
> |CLIP| 72.99|72.98|
> |DINOv2|87.13|85.15|
> |SigLIP|84.19|81.93|
>
>
> > Q3. F-ConV loss components are not ablated. Could the authors provide ablation studies isolating the shaping and consistency losses in F-ConV?
>
>
> Thanks for your constructive suggestion. We further conducted ablation experiments on this. The experimental results are shown in the table below.
>
> |                             | AUROC | AP    |
> |-----------------------------|-------|-------|
> | F-ConV                      | 93.77 | 93.38 |
> | F-ConV w/o Shaping Loss     |   89.17   |  87.89     |
> | F-ConV w/o Consistency Loss |    92.54   |    91.66   |
>
> > Q4. No analysis of failure cases is provided.
>
> We express our gratitude for the reviewer’s valuable suggestion. In the revised manuscript, we incorporate an analysis of failure cases as recommended. Our findings indicate that highly realistic generated images typically exhibit minimal feature variations pre- and post-transformation, resulting in misclassification.
>
> > Q5. The heavy mathematical formalism might limit accessibility.
>
> We express our appreciation for the reviewer’s valuable suggestion. In the revised manuscript, we incorporate more detailed and precise explanations to enhance clarity and accessibility.
>
>
> > Q6. How sensitive is the method to the choice of the transformation function $h(\cdot)$? Are some augmentations more effective than others?
>
> We thank the authors for the valuable suggestions. To address the query regarding the sensitivity of our method to the choice of transformation function $h(\cdot)$, we conducted additional ablation studies evaluating the effectiveness of various transformation functions. The results, presented in the table below, indicate that the three transformations examined exhibit comparable performance.
>
> |                         | AUROC | AP    |
> |-------------------------|-------|-------|
> | ConV                    | 87.13 | 85.15 |
> | ConV w/o HorizontalFlip | 86.55 | 84.19 |
> | ConV w/o ColorJitter    | 85.97 | 83.74 |
> | Conv w/o GaussianBlur   | 85.49 | 82.83 |
>
>
> > Q7. How is the performance better than a simpler setup where we train an autoencoder over training images and detect generated images if the loss is higher than a threshold?
>
> We express our appreciation for the reviewer’s insightful comment. The approach highlighted, a well-established method in anomaly detection, leverages reconstruction loss through autoencoders. These models are typically trained on normal samples, enabling effective reconstruction of in-distribution data but yielding higher reconstruction errors for anomalous samples. However, in the context of AI generated image detection, the diverse distributional characteristics of natural images necessitate training an autoencoder on an extensive dataset, such as the LVD-142M dataset employed in DINOv2, to adequately capture the natural image distribution. Due to resource and time constraints, training an autoencoder on such a large-scale dataset is infeasible within the scope of this study, precluding a fair comparison with this method.

---

> > ### Comment · Reviewer_3bZe · 2025-08-04
> >
> > Thanks to the authors for addressing the questions and providing additional experiments to support the proposed pipeline.
> >
> > Regarding the first question, in addition to the empirical results, it would be helpful if the authors could also offer some theoretical intuition or justification for the assumption of orthogonal subspaces of gradients.

---

> > > ### Author Response · Authors · 2025-08-05
> > >
> > > We appreciate the reviewer’s insightful comments, which prompted us to carefully validate the hypothesis proposed in the referenced study. For Q1, initially, we attempted to conduct experiments using the DINOv2 framework. However, as DINOv2 requires both teacher and student models to compute the loss, and the teacher model’s weights were unavailable, we were unable to proceed with this approach. Efforts to adapt alternative loss functions also proved unsuccessful. To address this, we pivoted to a robust experimental setup using a ResNet50 model pre-trained on ImageNet. Specifically, we utilized cross-entropy loss as the objective function and computed gradients with respect to both the original and augmented input samples. The cosine similarity between these gradients was then calculated to assess their divergence. Across 100 randomly selected samples, the cosine similarity ranged from 0.0005 to 0.017, demonstrating consistently low similarity. These findings provide strong empirical evidence supporting the theoretical framework proposed in our study.
> > >
> > >
> > > Intuitively, for the data point $\mathbf{x} _ {\mathcal{M}}$ on the natural manifold $\mathcal{M}$,  since the model has been well trained, the gradient of the loss function $\nabla f _ 1$ used in training is orthogonal to the tangent space of the manifold $\mathcal{T}(\mathcal{M})$ (otherwise, even a small perturbation in the input along tangent space will result in significant changes in loss.).
> > >
> > > This indicates that:
> > >
> > >  $\nabla f _ 1\left(\mathbf{x} _ {\mathcal{M}}\right) \in \mathcal{O}\left(\mathbf{x} _ {\mathcal{M}}\right)$.
> > >
> > >  For $\nabla f _ 2\left(\mathbf{x} _ {\mathcal{M}}\right)$, since $f_2:=f _ 1 \circ h$, according to the chain rule, we can split it into two terms:
> > >
> > >  $\nabla f _ 2\left(\mathbf{x} _ {\mathcal{M}}\right)=\mathbf{J} _ {h\left(\mathbf{x} _ {\mathcal{M}}\right)} \frac{\partial f _ 1\left(h\left(\mathbf{x} _ {\mathcal{M}}\right)\right)}{\partial h\left(\mathbf{x} _ {\mathcal{M}}\right)},$
> > >
> > >  where $\mathbf{J} _ {h\left(\mathbf{x} _ {\mathcal{M}}\right)}$ is the Jacobian matrix of $h\left(\mathbf{x} _ {\mathcal{M}}\right)$. $h(\cdot)$ is the transformation function, modeling the transformation along local data manifold, and $\mathbf{J} _ {h\left(\mathbf{x} _ {\mathcal{M}}\right)}$ models the tangent space at point $\mathbf{x} _ {\mathcal{M}}$. Therefore:
> > >
> > >  $\nabla f _ 2\left(\mathbf{x} _ {\mathcal{M}}\right)^{\top} \nabla f _ 1\left(\mathbf{x} _ {\mathcal{M}}\right)=\frac{\partial f _ 1\left(h\left(\mathbf{x} _ {\mathcal{M}}\right)\right)^{\top}}{\partial h\left(\mathbf{x} _ {\mathcal{M}}\right)} \underbrace{\mathbf{J} _ {h\left(\mathbf{x} _ {\mathcal{M}}\right)}^{\top}} _ {\text{tangent space}}  \underbrace{\nabla f _ 1\left(\mathbf{x} _ {\mathcal{M}}\right)} _ {\text{orthogonal space}}=0.$
> > >
> > >  We have incorporated the above detailed discussion and experimental verification results into the revised version.

---

> > > > ### Comment · Reviewer_3bZe · 2025-08-06
> > > >
> > > > Thanks to the authors for their reply. The reviewer considers that all raised concerns have been addressed.

---

### Comment · Area_Chair_hX6d · 2025-08-02
**Reviewer-Author Discussions**

Dear Reviewers,


Could you kindly review the authors’ rebuttal as well as the comments from your fellow reviewers, and share your thoughts on the authors’ responses? Many thanks.


Best regards,

AC

---

### Decision · Program_Chairs · 2025-09-17

**Decision:**

Accept (spotlight)

**Comment:**

This paper was reviewed by four experts in the field. The paper received all positive reviews, i.e., 4 Borderline Accept, 5 Accept, 4 Borderline Accept, 4 Borderline Accept.
After the discussion, most of the concerns raised by reviewers have been well addressed
Based on these positive reviews, the decision was made to recommend it for acceptance. We congratulate the authors on their acceptance!
Besides, authors should revise the paper taking into account the reviewers' comments.